# Modelling the effects of ice transport and sediment sources on the form of detrital thermochronological age probability distributions from glacial settings

Maxime. Bernard[1], Philippe. Steer[1], Kerry. Gallagher[1], David Lundbek. Egholm[2]

[1]Univ Rennes, CNRS, Géosciences Rennes, UMR 6118, 35000 Rennes, France.
[2]Department of Geoscience, Aarhus University, Aarhus, Denmark.

**Correspondence**: Maxime Bernard (maxime.bernard@univ-rennes1.fr)

**Abstract.** The impact of glaciers on the Quaternary evolution of mountainous landscapes remains controversial. Although in situ or bedrock low-temperature thermochronology offers insights on past rock exhumation and landscape erosion, the method also suffers from potential biases due to the difficulty of sampling bedrock buried under glaciers. Detrital thermochronology attempts to overcome this issue by sampling sediments, at e.g. the catchment outlet, a component of wich may originate from beneath the ice. However, detrital age distributions not only reflect the catchment exhumation, but also spatially variable patterns and rates of surface erosion and sediment transport. In this study, we use a new version of a glacial landscape evolution model, iSOSIA, to address the effect of erosion and sediment transport by ice on the form of synthetic detrital age distributions. Sediments are tracked as Lagrangian particles formed by bedrock erosion, and their transport is restricted to ice or hillslope processes until they are deposited. We base our model on the Tiedemann glacier (British Columbia, Canada), which has simple morphological characteristics, such as a linear form and no connectivity with large tributary glaciers. Synthetic detrital age distributions are generated by specifying an erosion history, then sampling sediment particles at the frontal moraine of the modelled glacier.

Results show that sediment sources, reflecting different processes such as glacier and hillslope erosion, can have distinct bedrock age distribution signatures, and estimating such distributions should help to identify predominant sources in the sampling site. However, discrepancies between the detrital and the bedrock age distributions occur due to (i) the selective storage of a large proportion of sediments in small tributary glaciers and in lateral moraines, (ii) the large range of particle transport times, due to varying transport lengths and to a strong variability of glacier ice velocity, (iii) the heterogeneous pattern of erosion, (iv) the advective nature of glacier sediment transport along ice streamlines. This last factor leads to a poor lateral mixing of particle detrital signatures inside the frontal moraine and then local sampling of the frontal moraine is likely to reflect local sources upstream. Therefore, sampling randomly across the moraine is preferred for a more representative view of the catchment age distribution. Finally, systematic comparisons between synthetic (U-Th)/He and fission track detrital ages, with different bedrock age-elevation profiles and different relative age uncertainties, show that the nature of the age-elevation relationship and age uncertainties largely control the ability to track sediment sources in the detrital record. However, depending on the erosion pattern spatially, qualitative first-order information may still be extracted from thermochronological

system with high uncertainties (> 30 %). Overall, our results demonstrate that detrital age distributions in glaciated catchments are strongly impacted not only by erosion and exhumation but also by sediment transport processes and their spatial variability. However, when combined with bedrock age distributions, detrital thermochronology offers a novel means to constrain the transport pattern and time of sediment particles.

## 1 Introduction

Glaciers have left a profound impact on the topography of mountainous landscapes in particular by eroding deep glacial valleys and depositing large volume of sediments in moraines. As glacier dynamics are linked to climate, an active area of research aims to characterize the role of glacial erosion on the dynamics and relief development of mountain belts during the recent Quaternary glaciations (e.g. Zachos et al., 2001; Molnar and England, 1990; Beaumont et al., 1992; Montgomery, 2002; Brozović et al., 1997; Whipple et al., 2009; Steer et al., 2012; Champagnac et al., 2014). To address these questions, two timescales have typically been considered: a longer timescale ($10^5$-$10^6$ years) to assess the potential glacial imprint on the landscape, and a shorter timescale ($10^1$-$10^4$ years) to understand how ice actually erodes the landscape. For example, some studies integrated glacial sediment records worldwide and in situ low-temperature thermochronology data, to estimate glacial erosion rates. They showed average erosion rates of $10^0$-$10^3$ mm yr$^{-1}$ on short timescales ($10^3$-$10^5$ years) and long-term ($>10^6$ years) average erosion rates of $10^{-2}$-$10^0$ mm yr$^{-1}$ (Hallet et al., 1996; Koppes and Montgomery, 2009; Valla et al., 2011b; Koppes et al., 2015; Bernard et al., 2016).

Herman et al. (2013) used a global compilation of in situ thermochronological data and an inverse approach to infer an increase in erosion rates for all mountain ranges in the Quaternary period. They suggested that this effect is more pronounced for glaciated mountains, implying a significant role of glaciation on erosion rates. However, the results of this study have been contested (Willenbring et al., 2016; Schildgen et al., 2018) and thus the conclusion debated. More recently, Yanites and Ehlers (2016) correlated high glacier ice sliding velocities with high denudation rates deduced by thermochronological data in the southern Coast Mountains, British Columbia. They also found that glacial erosion may only occurs for less than 20% of a glacial-interglacial cycle, in some areas, and may explain the discrepancy between the longer and shorter timescale for erosion rates. In situ thermochronology consists of collecting bedrock samples from discrete locations to map thermochronological ages and exhumation rates (e.g., Fitzgerald and Stump, 1997; Valla et al., 2011b, Herman et al., 2013). However, this approach offers a potentially biased assessment of catchment wide exhumation pattern as it is dependent on the spatial bedrock sampling strategy (Ehlers, 2005, Valla et al., 2011a). For example, the age-elevation relationship inferred from in situ or bedrock thermochronology data may not capture younger ages expected along the valley floor, and buried under the ice (Enkelmann et al., 2009; Grabowski et al., 2013).

In glacial environments, erosion mostly occurs subglacially through abrasion and quarrying (Boulton, 1982; Hallet et al., 1996) or supraglacially (i.e. ice-free areas) through periglacial mechanisms and gravitational processes (Matsuoka and Murton, 2008). These two sources (i.e. supraglacial and subglacial areas), and their contribution to erosion, define the relative

proportion of sediments that enter the glacier transport system (Small, 1987). Sediments are transported by glaciers by (i) subglacial water flow through cavity or channel systems (Kirkbride, 2002; Alley et al., 1997; Spedding, 2000), and (ii) by ice internal deformation for sediments incorporated within or above the ice (Hambrey et al., 1999; Goodsell et al., 2005).

Therefore, sampling and analysing these sediments with detrital thermochronology has the potential to provide a more spatially integrated view of the catchment erosion pattern on short timescales ($10^1$-$10^4$ years) and thereby a more representative indication of how ice is eroding the landscape (e.g. Stock et al., 2006; Tranel et. 2011; Ehlers et al., 2015). The detrital sampling protocol focusses on collecting glacial deposits, generally close to the outlet of the catchment (Ruhl and Hodges, 2005; Stock et al., 2006; Falkowski et al., 2016; Glotzbach et al., 2018). A major motivation for detrital sampling is that we can potentially obtain grains from regions inaccessible for bedrock sampling due to ice cover and lack of outcrop or, more pragmatically, logistical considerations (e.g. cost). However, a priori knowledges about the distribution of thermochronological ages with elevation (e.g. Brewer et al., 2003), and the mineral fertility of the sources (Moecher and Samson, 2006), are often required to reliably interpret the data from detrital thermochronology.

Thermochronological age distributions of detrital samples are generally presented as synoptic probability density functions (SPDF) (Brewer et al., 2003; Ruhl and Hodges., 2005). Analysis of many glaciated catchments has been based on the interpretation of these SPDFs (Stock et al., 2006; Tranel et al., 2011; Avdeev et al., 2011; Thomson et al., 2013). More recently, Enkelmann and Ehlers (2015) investigated the degree of mixing of sediments from ice cores at the terminus of the Tiedemann glacier (Coast Mountains, Canada) using both detrital and in situ low-temperature thermochronology. Comparison of the detrital age distributions from the ice-cored terminal moraine and from glacial outwash (Ehlers et al., 2015), showed that sampling through the terminal moraine is similar to sampling the proglacial river, supporting the idea of an efficient vertical mixing of glaciated sediments at the glacier front. The latter result is also consistent with results obtained on the Malaspina Glacier, Alaska (Enkelmann et al., 2009; Grabowski et al., 2013). However, Enkelmann and Ehlers (2015) pointed out the sensitivity of their results to the relatively high uncertainties on the ages. Furthermore, while the shape of detrital SPDFs is expected to be mainly controlled by the catchment exhumation history, other processes such as grain erosion, transport and deposition are likely to play a role. Consequently, characterizing the effect of surface processes on the shape of detrital SPDFs may help improve their interpretation.

In this study, we present a numerical approach that allows us to explore the effect of sediment transport by ice and the role of different source areas (i.e. subglacial and supraglacial) in shaping of detrital SPDFs at a glacier front. To this end, we combined a new version of a glacial landscape evolution model, iSOSIA (Egholm et al., 2011) that allows tracking of sediments, with an external routine that calculates the detrital thermochronological age distributions. We chose to apply our numerical approach on the Tiedemann glacier catchment, as it shows simple morphological characteristics and because thermochronological data are available (Enkelmann and Ehlers, 2015; Ehlers et al., 2015). We produced our bedrock synthetic thermochronological data from two low-temperature thermochronological systems, apatite (U-TH)/He and apatite fission track (i.e. AHe and AFT respectively), according to the age-elevation profiles from Enkelmann and Ehlers (2015) for the AFT ages and from Ehlers et al. (2015) for the AHe ages. First, we investigate the kinematics of our sediment transport model in the modelled catchment,

to assess the role of the ice transport. Next, we focus on the role of the sediment sources by calculating detrital SPDFs resulting from uniform catchment erosion across the catchment, as well as, hillslope-only sources and glacial-only eroded sources. Finally, we consider non-uniform catchment erosion, by letting iSOSIA and its implicit erosion laws control the pattern of erosion. This last experiment allows us to assess the record of complex erosion processes potentially contained in detrital SPDFs. For all experiments, the sediments are transported by ice and hillslopes, so that transport by the hydrological system is neglected. The two thermochronological methods (i.e. AFT and AHe) considered, have different thermal sensitivities and thus, different spatial distributions of bedrock ages in the landscape. They typically have different age uncertainties allowing us to address the effect of these uncertainties on the interpretation of SPDFs.

## 2 Methods

### 2.1 The glacial landscape evolution model: iSOSIA

iSOSIA is a one-layer depth-integrated second order shallow ice approximation model (Egholm et al., 2011). It is able to simulate ice flowing on relatively steep topography, through the computation of membrane stresses (Hindmarsh, 2006) and with a depth-integrated ice flow velocity. These two characteristics make iSOSIA more accurate than models based on the zeroth-order shallow ice approximation, while being computationally efficient compared to models that resolve the full 3D set of Navier-Stokes equations (Elmer/Ice, Braedstrup et al., 2016). A full description of iSOSIA and the relevant ice flow equations can be found in Egholm et al. (2011, 2012a, 2012b) and Braedstrup et al. (2016).

Glacial hydrology plays an important role on the ice sliding velocity (Clarke, 1987), and has been recently investigated in numerical models (Schoof, 2005, 2010; Iverson, 2012; Ugelvig et al. 2016; Ugelvig et al. 2018). Here we adopt a simplified version of the sliding law of Schoof (2005), that accounts for the opening of cavities due to the roughness of the bed. This defines the area on which the basal shear stress ($\tau_b$) is applied as well as the basal sliding speed ($u_b$), as below

$$u_b = C_s . \frac{\tau_b^n}{C}, \tag{1}$$

where $C_s$ is the ice sliding constant and $C = 1 - \frac{S}{L}$ defines the proportion area on which $\tau_b$ is applied, where L is the length between two topographic steps (Fig. S1). The steady-state length of cavities (S) is controlled by the effective mass balance (i.e. accounting for basal and internal melting of the ice) that determines the annual average water flux over the glacier, $\mathbf{q_w}$. This is specified as follows

$$S = \left( \frac{L_s . q_w}{k_0 . \sqrt{\nabla \psi}} \right)^{0.8} . \frac{1}{\beta . h_s}, \tag{2}$$

where $L_s$ is the mean cavity spacing in a cell (see Table 1 for all parameter values), $k_0$ is the minimum hydraulic conductivity, $\nabla \psi = \rho_w g \nabla h_{ice}$ the hydrological head, $\beta$ is a scaling factor, and $h_s$ is the mean height of the topographic step and depends

**Table 1:** List of parameters described in the text.

| Parameters | Description | Value/Unity |
| --- | --- | --- |
| **Climate** | | |
| $T_{sl}$ | Sea level temperature | 9° C |
| $dT_{air}$ | Lapse rate | $0.006$°C m$^{-1}$ |
| $m_{acc}$ | Accumulation gradient | 0.08 m.yr$^{-1}$ °C$^{-1}$ |
| $m_{abl}$ | Ablation gradient | 0.25 m.yr$^{-1}$ °C$^{-1}$ |
| **Ice** | | |
| n | Ice creep constant | 3.0 |
| B | Ice viscosity constant | 73.3x10$^6$ Pa s$^{1/3}$ |
| Cs | Ice sliding constant | 1.29x10$^{-2}$ Pa s$^{1/3}$ |
| $h_{ice}$ | Ice surface elevation | Model outcome |
| H | Ice thickness | Model outcome |
| $\nabla b_s$ | Bed slope in direction of ice flow | Dimensionless |
| $k_q$ | Quarrying coefficient for erosion | 1.97x10$^{-4}$ s Pa$^{-3}$ |
| **Hillslope** | | |
| $S_c$ | Critical slope | 1.4 |
| $K_h$ | Hillslope diffusivity | 5 m$^2$y$^{-1}$ |
| $k_{eh}$ | Hillslope erosion coefficient | 3.95x10$^{-5}$ m$^2$ y$^{-1}$ |
| **Particles** | | |
| $H_s$ | Particle sediment thickness | 0.01 m |
| $z_p$ | Burial depth of particles relative to the ice surface | Dimensionless |
| **Hydrology** | | |
| N | Effective pressure | Model outcome |
| $h_s$ | Bed step height | Model outcome |
| S | Cavities size | Model outcome |
| L | Bed step length | Model outcome |
| $L_s$ | Cavity spacing | 4 m |
| $k_0$ | Minimum hydraulic conductivity | 3.10x10$^{-4}$ kg$^{-1/2}$ m$^{3/2}$ |
| $q_w$ | Water flux | Variable |
| $\nabla\psi$ | Hydrological gradient | $\nabla\psi = q_w g \nabla h_{ice}$ |
| β | Cavity shape parameter | 0.7 |

linearly on the slope in direction of sliding (see Ugelvig et al., 2016; Eq. 10). The term in parentheses on the right-hand side of Eq. (2) represents the cross-sectional area of a cavity ($A_s$). The steady cavity size controls the effective pressure (N) of the system and influences the basal sliding speed. However, due to how cavities dictate the volume of water stored at the bed, the effective pressure is also influenced by the opening/closure rate of cavities following

$$N = B \left(\frac{8}{\pi}\right)^{\frac{1}{n}} \left[\frac{u_b h_s - \beta h_s \frac{\partial S}{\partial t}}{S^2}\right]^{\frac{1}{n}}, \tag{3}$$

where B and n are the ice creep parameters. This hydrological model is similar to that used in Ugelvig et al. (2016), and we refer the reader to this study for more details. We calibrated our hydrological model by a trial-error process by varying $k_0$ (Eq.2) to produce a reasonable value of basal ice sliding velocities (Table S1).

In iSOSIA, the mass balance $M$ is linearly proportional to the mean annual air temperature $T_{air}$:

$$M = \begin{cases} -m_{acc} T_{air}, & if\ T_{air} \leq 0 \\ -m_{abl} T_{air}, & if\ T_{air} > 0 \end{cases}' \tag{4}$$

where $m_{acc}$ and $m_{abl}$ are the mass accumulation and the ablation gradients with respect to temperature, respectively. $T_{air}$ is assumed to decrease linearly with elevation $h$ with a lapse rate $dT_{air}$ as

$$T_{air} = T_{sl} + dT_{air}.h, \tag{5}$$

where $T_{sl}$ is the sea level temperature set constant at 9°C.

As mentioned in the introduction, the two main mechanisms of glacial erosion are abrasion and quarrying (Boulton, 1982; Hallet et al., 1996). As debris resulting from abrasion is mostly small in size (i.e. <50 µm, Hallet, 1979), we follow MacGregor et al. (2009) and consider that this debris is instantaneously transported away from the glacial catchment by meltwaters. However, quarrying acting on the lee side of bedrock steps by plucking can produce larger debris. Therefore, we consider in this study that sediments are only produced by quarrying. Quarrying is generally accounted for by a classical basal sliding power-law, modified to account for some factors directly related to plucking (MacGregor et al. 2000; Kessler et al., 2008; MacGregor et al., 2009, Egholm et al., 2009; Herman et al., 2011). These factors include pre-existing fractures in the bedrock, effective pressure, the regional bed slope in direction of ice flow, variation in meltwater drainage and the ice-bed contact area determined by the opening of cavities (Iverson, 2012; Cohen et al., 2006; Ugelvig et al., 2016; Anderson, 2014; Ugelvig et al., 2018). We follow Ugelvig et al. (2016) in modelling the quarrying rate $E_q$ as

$$E_q = k_q N^3 u_b (\nabla b_s + b_{s0})^2, \tag{6}$$

where $k_q$ is a scaling coefficient for the effective pressure, N, that represents the effect of bedrock lithology and fractures, $\nabla b_s$ is the bed slope in direction of sliding and $b_{s0}$ is a term to allow for negative bed slopes. As we focus on glacier-induced erosion, we ignore fluvial erosion. However, we do consider hillslope erosion, which represents the supraglacial source of sediments. We assume that hillslope erosion, $E_h$, depends non-linearly on the bed gradient, as defined in (Andrew and Bucknam, 1987) and (Roering et al., 1999):

$$E_h = -\frac{k_{eh}.\nabla b}{1 - \left(\frac{|\nabla b_{ij}|}{S_c}\right)^2}\frac{dt}{dl}, \tag{7}$$

where $k_{eh}$ is an erodibility constant, $S_c$ the critical slope, dt the time step and dl the resolution of a cell (dl = 100 m).

## 2.2 Particle tracking and sampling

To calculate the detrital age distributions produced from a glaciated catchment, we need to track sediments from the source to

their final site of deposition. Accordingly, we define "particles" as sediment trackers in our models. Sediments are produced as a result of local erosion. When sediment thickness exceeds a threshold $H_s = 0.01$ m, a particle is generated and can be transported away from its initial cell. This threshold is chosen to lead to a reasonable number of particles within models while keeping a sufficient time resolution in the generation of particles (i.e. with an erosion rate of 1 mm yr$^{-1}$ and $H_s = 0.01$ m a particle is generated every 10 years in a cell). Depending on its position within the catchment, a particle is transported by

different processes. Similar to Eq. (7), we assume that the flux, $\mathbf{q}_{ph}$, of an unglaciated particle depends non-linearly on the bed gradient ($\nabla b$):

$$q_{ph} = -\frac{K_h.\nabla b}{1 - \left(\frac{|\nabla b|}{S_c}\right)^2}, \tag{8}$$

where $K_h$ is the hillslope diffusion coefficient.

Glaciated particles can be incorporated into the ice either from the ice surface, for supraglacial debris, or from the ice-bed

interface by basal quarrying. The horizontal velocity of each glaciated particle is computed as a function of their depth within the ice $z_p$, relative to the ice surface. Because iSOSIA is a one-layer depth-integrated model, the horizontal velocity depth-profile is computed following Glen's flow law (Glen, 1952). We follow Rowan et al. (2015) in assuming that, due to the ice viscosity, the horizontal ice velocity decays as a fourth-order polynomial of the ice thickness:

$$u_p(z_p) = \frac{5}{4}[1 - z_p^4]\bar{u} + u_b, \tag{9}$$

where $\bar{u} = \bar{u}_d + u_b$ is the average velocity of the ice, with $u_b$ the basal ice speed and $\bar{u}_d$ the average velocity due to the internal deformation of the ice which is approximated as a tenth-order polynomial of the local ice thickness, the bed and ice surface gradients, as well as its curvature (Egholm et al., 2011). We stress again that we neglect the transport of particles by meltwater and focus only on the ice and hillslope processes. Velocity profiles for glaciated and unglaciated particles are depicted in Fig. (S2).

The burial depth of particles ($z_p$) within the ice is expressed relative to the ice surface (i.e. $z_p = 0$ at the ice surface and $z_p = 1$ at its base, Fig. 1), and computed according to the mass balance:

$$\frac{\partial z_p}{\partial t} = \frac{z_p \dot{M}_b - (1 - z_p)\dot{M}_s}{H}, \tag{10}$$

where $\frac{\partial z_p}{\partial t}$ is the change in burial depth of a particle through time, $\dot{M}_b$ and $\dot{M}_s$ are the surface and basal melting at time t, respectively, and H is the ice thickness. Relative to the ice surface elevation, particles are moved downward according to the accumulation rate (in the accumulation zone $\dot{M}_s < 0$ ), or upward with the lowering of the ice surface due to melting (in the ablation zone $\dot{M}_s > 0$, Fig. 1).

In the experiments presented in the following sections, detrital particles are sampled throughout the frontal moraine, defined as the glacier tongue (Fig. 1 and Fig. 5d). As vertical mixing of sediment has been reported in some glacier fronts (Enkelmann et al., 2009; Grabowski et al., 2013; Enkelmann and Ehlers, 2015), we sample particles independently of their vertical position within the ice thickness to infer detrital age distributions.

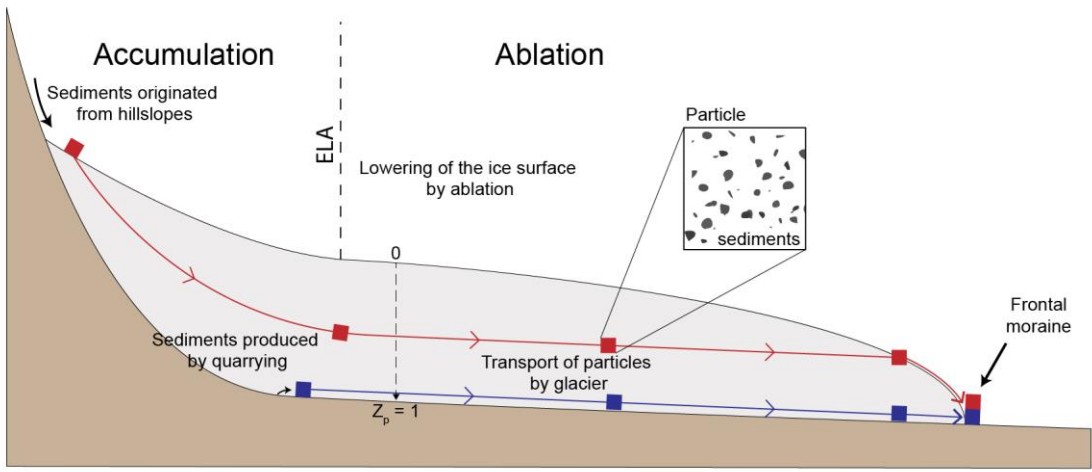

**Figure 1:** Particle transport by ice. Particles originated from hillslopes enter to the glacier system from the surface ($z_p = 0$), whereas particles produced by the ice erosion are incorporated in the basal ice ($z_p = 1$). The transport of glaciated particles follows the ice flow pattern from the sources to the frontal moraine. ELA is the equilibrium line altitude.

**2.3 Modelling the Tiedemann glacier**

Our main goal is to investigate the influence of sediments sources and transport by ice on the shape of detrital SPDFs at the glacier front. We present our modelling approach using the Tiedemann glacier catchment (Coast Mountains, British Columbia, Canada, Fig. 2) as a reference case. This region has been previously studied using detrital and in situ thermochronology (Ehlers et al., 2015; Enkelmann and Ehlers, 2015). Moreover, the Tiedemann glacier has simple morphological characteristics with a

linear main glacial valley and small-extent tributary glaciers. We calibrated our iSOSIA model mimicking the Tiedemann glacier, by a trial-error process. We varied climatic, ice and hydrological parameters (dTa, Cs, $M_{acc}$ and $k_0$, see Table S1) and compared the resulting mean ice thickness and location of glacier tongue against the results of the ITMIX experiments (Farinotti et al., 2016; Farinotti et al., 2019) that predicts thickness of many glaciers around the world, and thus the Tiedemann glacier (Fig. S3). To produce synthetic detrital age distributions resulting from this model, we restrict the glacier to be in

steady-state and impose a constant topography (i.e. erosion produces particles but no topographic changes). We stress that our goal is not to fit the dynamics of the actual Tiedemann glacier, which is in a phase of retreat (Tennant et al. 2012). Rather, we ask how surface processes affect the shape of detrital SPDFs in a simple case of constant and steady transport and erosion within a glaciated catchment.

The elevation of the Tiedemann glacier catchment ranges from 363 to 3920 m, for a total local relief of 3557 m. The resulting maximum ice thickness is 454 m with a mean ice velocity of 13.2 m yr$^{-1}$, and a range of 0 to 109.2 m yr$^{-1}$ (Fig. 3), which are

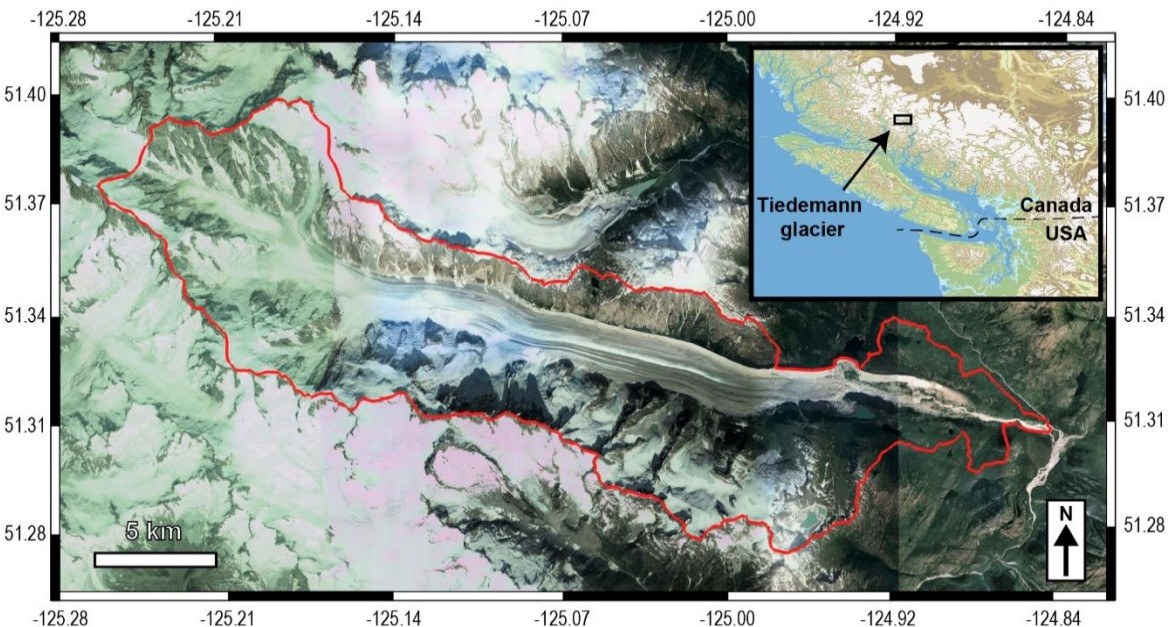

**Figure 2:** Satellite view of the Tiedemann glacier, British Columbia, Canada. The red contour represents the catchment area. The inset shows the regional location, in the Coast Mountain range, Canada. Note the dominantly linear pattern of sediments transport (image from ©Google Earth, Maxar Technologies; inset map with data from Jarvis et al., 2008).

characteristic of Alpine glaciers (Benn and Evans, 2013). The mean ice velocity is mainly controlled by the basal sliding speed, which in turn is influenced by the shear stress on the bed and the distribution of the effective pressure (Fig. S4). This results in two main regions of relatively high velocities (~60 m yr$^{-1}$) located at ~7 km and ~14 km in the glacier latitudinal coordinates

(Fig. 3c).

### 2.4 Synthetic ages and production of detrital age distributions

To produce detrital SPDFs we need to set the bedrock or in situ age distributions. To this end, we consider the age-elevation profiles from two low-temperature systems presented in Enkelmann and Ehlers (2015) (AFT ages), and from Ehlers et al. (2015) (AHe ages, Fig. 4a). We assume that true AHe and AFT ages in the bedrock at a given elevation are given by the age-

elevation relationship. In practice, thermochronological analysis on a single-grain is presented as an age and its associated uncertainty. Thus, the measured age can be considered as a noisy sample of the true age. To produce a SPDF, we simulate this

process for the detrital grain ages by adding random noise to the true ages, and assume the detrital grain ages are not modified during transport and deposition.

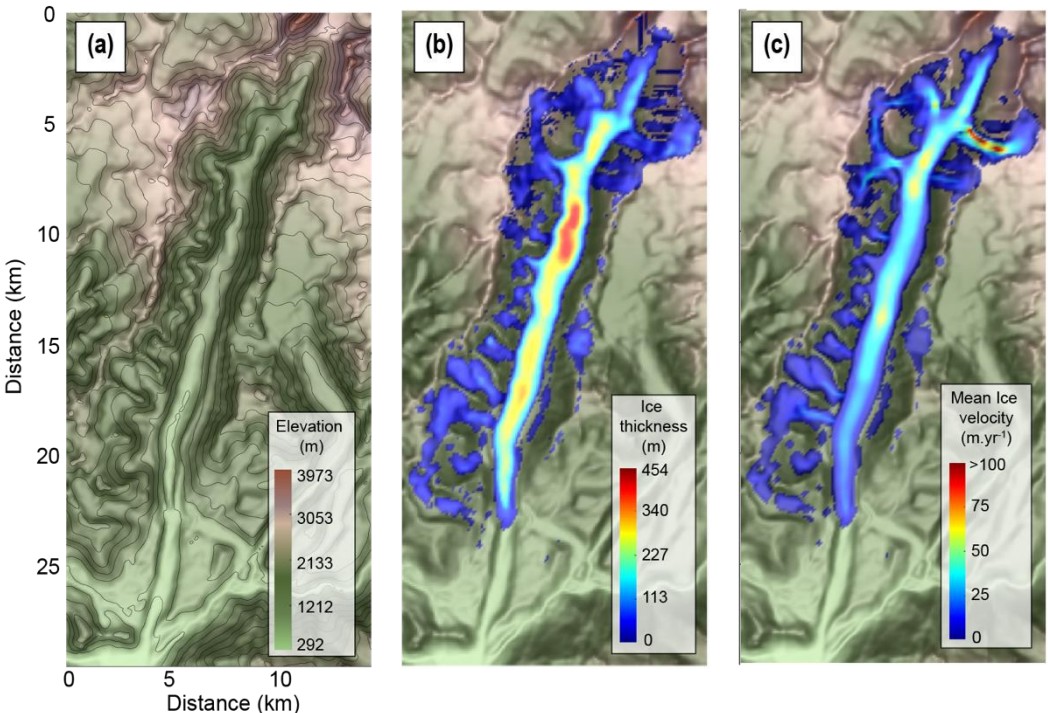

**Figure 3:** Characteristics of the Tiedemann glacier model. (a) The underlying topography, (b) the ice thickness, and (c) the mean ice velocity. The mean ice thickness has been calibrated with the ITMIX experiments (see Fig. S3)

In Ehlers et al. (2015), AHe bedrock ages and thus detrital data have relatively high uncertainties (~21%). However, we stress here that one of our goals is to assess the effect of age uncertainty on the interpretation of SPDF, then we chose to apply lower
uncertainty for the synthetic AHe ages. For this reason, we adopt uncertainties of 10% for the synthetic AHe ages sampled from the age-elevation profile. This is similar to the typical reproducibility in aliquots of single grain ages (e.g. ~6% Farley and Stocki, 2002).

The AFT data from Enkelmann and Ehlers (2015) show >30 % relative uncertainties. We generate synthetic AFT ages following an approach similar to Gallagher and Parra (2020). We use the external detector method (EDM) age equation (e.g.
Hurford and Green, 1983) to calculate the ratio of the spontaneous and the induced track density ($\rho_s/\rho_i$) corresponding to a given AFT age in the age-elevation profile. To allow for the effect of variable high uncertainties, we simulate the range of relative error of the Enkelmann and Ehlers (2015) study by randomly sample the published values of $N_s + N_i$ (i.e. the sum of the number of spontaneous and induced tracks). We then use a binomial distribution with parameter $\theta = \frac{\rho_s/\rho_i}{1+\rho_s/\rho_i}$ to sample $N_s$ values, corresponding to the number of spontaneous tracks, given the sampled value of $N_s + N_i$ (see Gallagher, 1995; Galbraith,
2005). With the randomly sampled value of $N_s$, conditional on the sampled value of $N_s + N_i$, we easily obtain a value for $N_i$. We can estimate a "noisy" AFT age and relative error from the sampled $N_s$ and $N_i$ values by using the EDM equation. For both

the AHe and AFT systems, we attribute for each particle a single age and uncertainty in accordance with its source location elevation.

The probability age distributions are then produced by using Gaussian kernel density estimate (Brandon, 1996; Ruhl and Hodges, 2005) with the mean and standard deviation equal to the age and error for both the AHe and AFT ages, following

$$SPDF_k = \frac{1}{N_p} \frac{1}{\alpha\sigma_{t_k}\sqrt{2\pi}} exp\left[-\frac{1}{2}\left(\frac{t - t_k}{\alpha\sigma_{t_k}}\right)^2\right], \tag{11}$$

where $N_p$ is the total number of sampled particles (i.e. ages), t is the range of ages in which we compute the SPDF, $t_k$ the age of a particle considered, $\sigma_t$ the associated age uncertainty, and $\alpha = 0.6$ is a scaling factor that handles the resolution and the precision of age components in a SPDF (Brandon, 1996). To compare age distributions, we also define a cumulative synoptic probability density function, which is simply the cumulated sum of the SPDF:

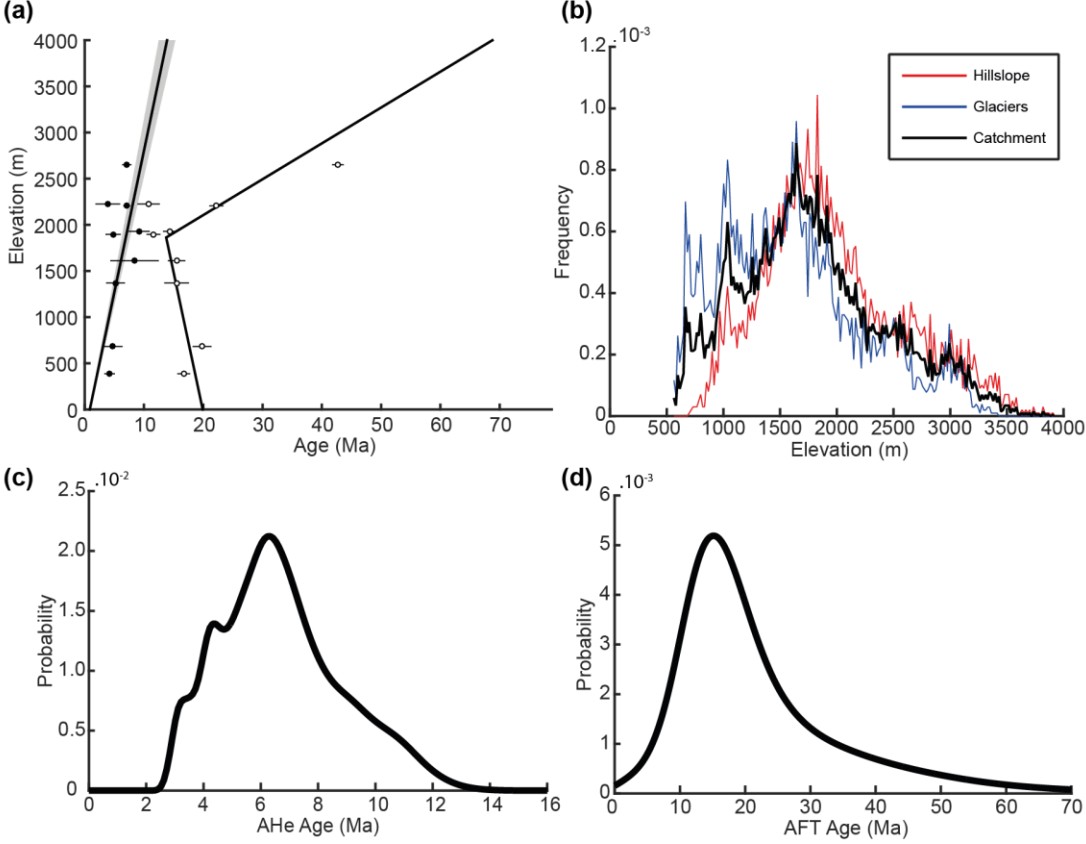

**Figure 4:** Synthetic bedrock thermochronological age distributions for the Tiedemann glacier model. The age-elevation profile for the two thermochronological systems AHe and AFT (a). Black and white circles represent the bedrock AHe and AFT ages from Ehlers et al., (2015) and Enkelmann and Ehlers (2015) respectively. Solid black lines are the best fit solution for the data. Shaded area for the AHe age-elevation show the 10% uncertainty considered for AHe ages. The catchment hypsometry frequency as well as glaciers and hillslope hypsometry frequencies are presented in (b). The bedrock AHe distribution considering 10 % relative uncertainties and bedrock AFT distribution with >30% relative uncertainties are shown in (c) and (d) respectively.

$$CSPDF = \sum_{j=0}^{N_p} SPDF_j \qquad (12)$$

Given the relatively high number of particles within the frontal moraine (i.e. > $10^6$) and to mimic real detrital sampling, all the detrital SPDFs presented in the following sections are produced by randomly sampling 105 particles from the total, independently of their source origin (Sections 3.2, 3.3 and 3.5) or accordingly (Section 3.4). This is similar to the proposed

minimum number of ages to adequately resolve a component representing >5% of the total detrital population (Vermeesch, 2004). We repeat this sampling process 10,000 times, storing the resulting SPDFs. From all of these detrital SPDFs, we compute the mean detrital SPDF and present the range of inferred detrital distributions.

## 3 Results

### 3.1 Ideal detrital age distributions

The bedrock age distributions for the two thermochronological systems, AHe and AFT (Fig. 4c-d) represent the ideal detrital age distributions if we sample the total catchment uniformly (allowing for the added noise). In the subsequent text, we refer to these ideal detrital age distributions as the bedrock distributions. The bedrock AHe SPDF shows a major peak ∼6.3 Ma with two youngest smaller peaks ∼3.3 and ∼4.3 Ma, and an older one ∼11 Ma. The shape of this SPDF mimics the hypsometric curve (black curve, Fig. 4b) which is expected with a uniform production of sediments and a linear age-elevation relationship.

The AFT SPDF shows a single major peak ∼16 Ma and the form does not mimic the hypsometric curve. This reflects partly the form of the age-elevation relationship, in which the ages below 2000 m (Fig. 4a) are similar, limiting the ability to resolve a unique relationship of age and elevation. We also note that, due to the high uncertainty, the probability to have an AFT age equal to zero is non-null.

### 3.2 Transport of detrital particles

Here, we investigate the kinematics of particles in our reference model for the Tiedemann glacier at the equilibrium state. To this end, we produce one particle in each cell of the model simultaneously (Fig. 5a) and let particles to be transported away potentially to the glacier front (i.e. the frontal moraine) where they would be deposited (Fig. 5b-d). In this case, assuming we have reached steady state in terms of transport, differences between the final detrital and the bedrock SPDFs can only be related to the process of particle transport.

We investigate the evolving contribution of particle source locations to the frontal moraine, and to the inferred detrital age distributions (Fig. 6). Particles are sampled independently of their source origin (hillslope vs glacial, Fig. 5a). According to our model, the minimum time required for a particle originated from elevated parts of the catchment (i.e. far from the glacier

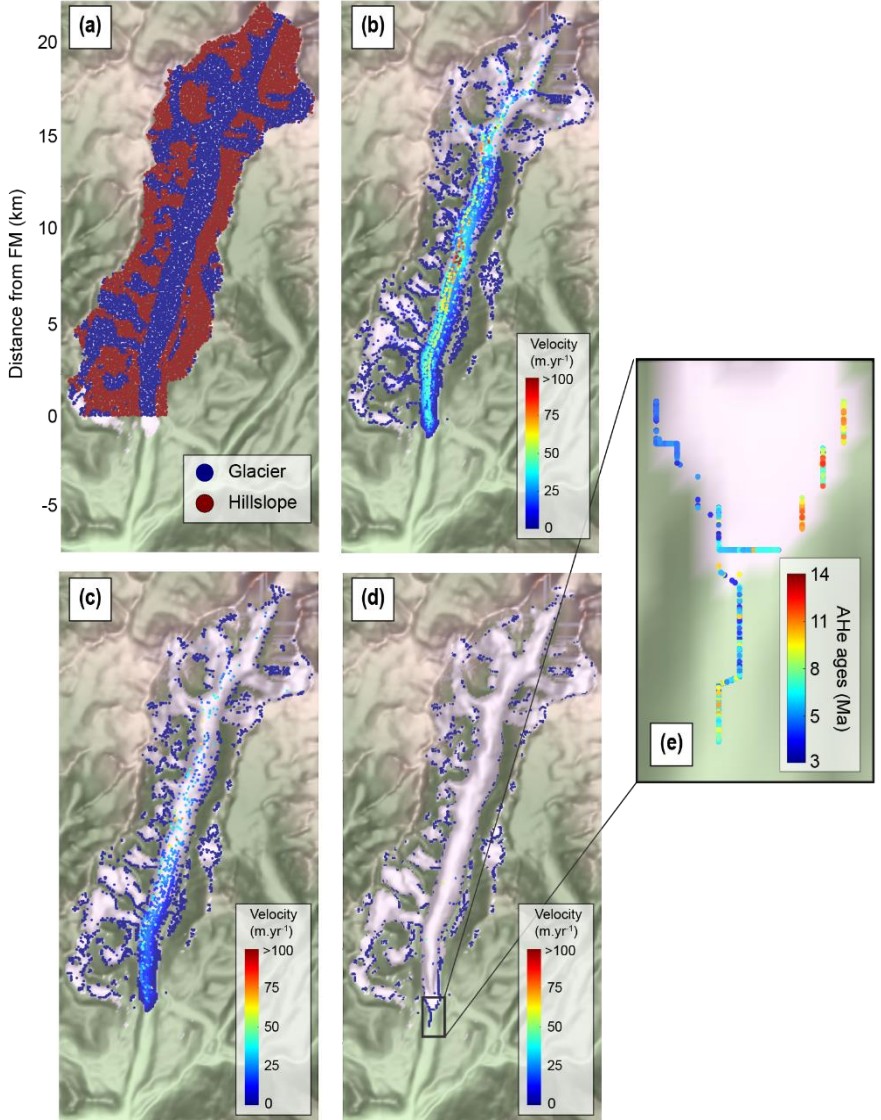

**Figure 5:** Kinematics of particles transport. Particles have been uniformly produced across the catchment and transported. (a) Sources of particles, red dots and blue dots are particles originated from hillslope and glacier at the beginning of the model simulation, respectively. (b) Particles velocity at time = 500 yr, (c) at time = 1000 yr, and (d) at time = 8500 yr. AHe ages in the frontal moraine (FM) are shown in (e) and see inset in (d).

front) to reach the frontal moraine, is ~500 years (Fig. 6a, pink curve). Furthermore, the proportions of particle source locations in the frontal moraine becomes nearly constant after ~1500 years. After 8500 years, some particles are still close to their sources (Fig. 5d). We find that only 44% of the initial particles have reached the frontal moraine (Fig. 6b, and Fig. S5a). The other 56% is trapped upstream, with a large part of them resting in a lateral moraine (Fig. 5d and 6b) or on the side of small

5   tributary glaciers having very low velocities (i.e. < 1 m yr$^{-1}$, Fig. 3c). These low velocities seem to result from a morphodynamic

feedback as the slope of tributary glaciers that carry the remaining particles is close to zero in the direction of sliding (Fig. S5b). This could, in turn, be related to the location of the ELA (Fig. 6b). As the number of particles within the frontal moraine does not increase significantly after 2000 years of simulation (i.e. reaching around 37% of the total number of initial particles, see Fig. S5a), we consider the detrital SPDF to have reached steady-state by this time.

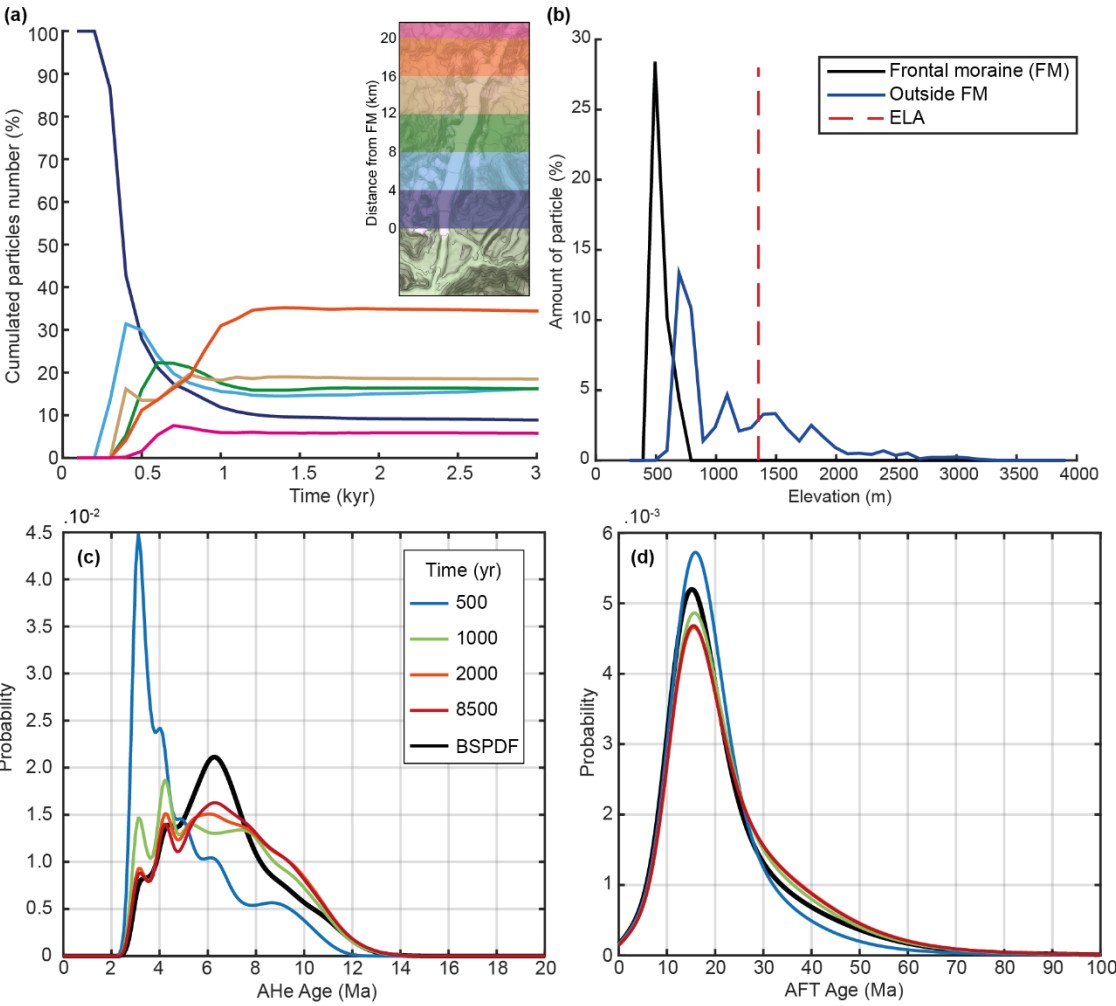

**Figure 6:** Transient evolution of particle sources and detrital thermochronological age distributions in the frontal moraine. (a) Cumulative contribution of source locations for particles, (b) their distribution with elevation after 8500 years of transport. (c) and (d) show the transient evolution of the detrital thermochronological age distribution in the frontal moraine for AHe and AFT systems respectively. BSPDF: Bedrock SPDF.

We observe a relative decrease of young ages (blue curves) in the detrital SPDFs (Fig. 6c-d) with time during the transient phase, reflecting the progressive arrival at the frontal moraine of older ages coming from upstream and elevated parts of the catchment (Fig. S6). The shape of detrital SPDFs reaches a steady-state around 2000 years of simulation.

The final form of the AHe detrital SPDF (i.e. red curve), shows a lower age peak ~6.3 Ma and a larger proportion of ages between 8 and 12 Ma compared to the bedrock SPDF. The two youngest age peaks, ~3.3 and ~4.3 Ma, in the detrital SPDF are close to those of the bedrock SPDF. Thus, the results suggest that the ice transports and deposits mid-range ages, around 6 Ma (i.e. ~1500-2000 m), higher in the catchment.

For the AFT system (Fig. 6d), the magnitude of age peak ~16 Ma is also lower than that of the bedrock age distribution. The relative contribution of old ages (> 25 Ma) is a little higher in the detrital SPDF, but for ages younger than 10 Ma, the distributions are similar. The differences suggest that ice also transports and deposits sediments mostly originating from 1000-2000 m in this case. We note that, the transient detrital SPDF at 1000 years (green curve, Fig. 6d) best matches the bedrock curve.

## 3.3 Detrital signature under uniform catchment erosion

Our model shows that more than half of the particles may not reach the frontal moraine. Consequently, we want to assess how the discrepancy between the detrital age distribution and the bedrock SPDF evolves if we consider a continuous production of particles (i.e. uniform erosion) across the catchment. To this end, we force the erosion rate to be 1 mm yr$^{-1}$ in each cell of the Tiedemann glacier catchment (Fig. 7a). This is within the range of natural values and it allows a continuous production of particles while maintaining a reasonable simulation time (i.e. a particle is produced every 10 years in each grid cell). In this section, no distinction is made between the two sources in the sampling process when building of detrital SPDFs. We stop the simulation after 2000 years as the detrital SPDF is close to equilibrium (Fig 6), and because the total number of particles increases rapidly which increases the computational time. Here, we focus our analysis on the AHe system for clarity but the detrital SPDFs for the AFT system can be found in the supplementary material (Fig S7).

First, we observe a large variability of detrital SPDFs resulting from sampling 105 particles in the frontal moraine (Fig. 7c). However, the bedrock age distribution is always included in the range of inferred detrital SPDFs. Second, the mean detrital SPDF (red curve) represents the "true" detrital signal we expect to obtain by considering all the particles in the sampling site. This mean detrital SPDF differs from the previous model (Fig. 6c), in that the younger ages are over-represented with two high peaks (at ~3.3 and ~4.3 Ma); and also, there is an excess of older ages (8-11 Ma). However, mid-range ages ~5 to ~8 Ma are still under-represented compared to the bedrock SPDF. Looking at the cumulative probability age distributions (CSPDF), differences between the detrital CSPDF and the bedrock CSPDF are less clear, with the most obvious differences between 3 and 6 Ma.

The detrital AFT SPDFs (Fig.S7) show a similar tendency in under-representing mid-ages (~16 Ma) but much less pronounced, with a maximum difference with the bedrock SPDF at ~12 Ma and a smaller one at ~28 Ma. The AFT bedrock and detrital CSPDFs have a maximum difference at ~19 Ma, but are similar overall. The results show that detrital SPDFs are shifted

toward younger ages, and deviate from what is expected for the case of uniform sampling (i.e. we expect the bedrock SPDF). We assign this behaviour to the short total transport distance for such ages. Furthermore, the SPDF is more informative on the effect of transport than the CSPDF that tends to smooth the signal. Finally, the quality of the results depends on the age uncertainty of the thermochronological system and on the age-elevation profile.

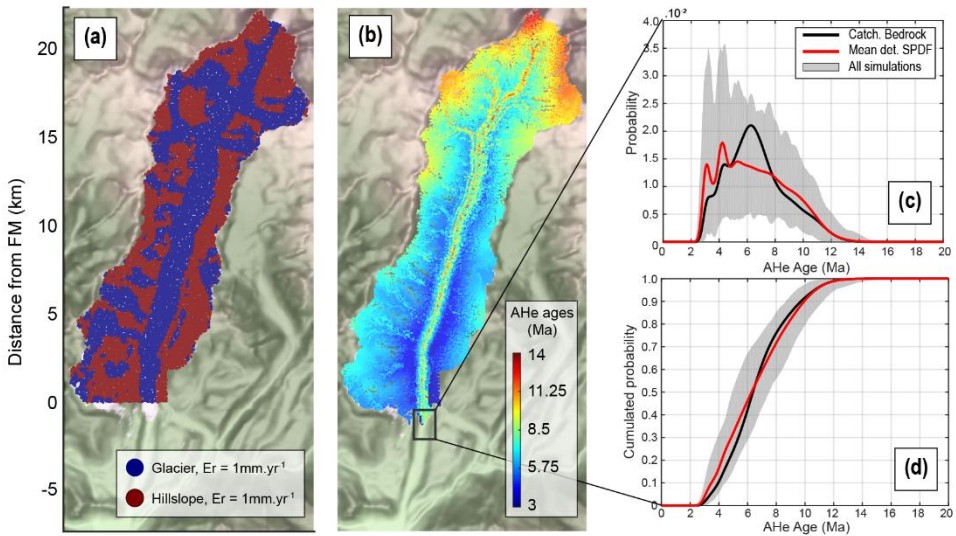

**Figure 7:** Spatial distribution of particles and detrital AHe age distributions within the frontal moraine. (a) Source locations of particles with Er being the erosion rate. The spatial AHe age distributions of particles after 2000 yrs of transport (b). The detrital AHe age probability distributions (c) and their cumulatives (d). The grey shaded area represents the range of the inferred 10,000 detrital SPDFs from the sampling process. The black square in (b) shows the frontal moraine (FM) position.

### 3.4 Detrital signature of glacier and hillslope sources

To understand the form of the mean detrital SPDF above, we now assess the respective influence of the two main sources of eroded particles, the ice-free hillslopes and glaciers. We first consider the ice-free hillslope sources (Fig. 7a). Within the frontal moraine we sample only particles that originated from hillslopes. Thus, the detrital AHe ages range from 3.1 to 13.8
10  Ma spanning ~96 % of the total age range shown by the age-elevation profile (i.e. from 2.7 to 13.8 Ma). We calculate a new bedrock SPDF according to the hypsometric distribution of hillslope sources and the AHe age-elevation profile (Fig. 4). The range of inferred detrital SPDFs is lower than the previous case (i.e. grey shaded area) but still remains important. The mean detrital SPDF shows a plateau-like signal from ~5 to ~9.5 Ma, suggesting that elevations from ~1400 to ~2500 m contribute in the same proportion to the particle budget in the frontal moraine. However, a comparison with the hillslope bedrock SPDF
15  indicates that storage of sediment occurs for such elevations. Moreover, the hillslope mean detrital SPDF is shifted toward older ages (> 8 Ma) and excludes ages lower than 3.5 Ma as hillslope erosion does not operate at the lower elevations which have these young ages (Fig. 4b). The hillslope mean detrital CSPDF shows a maximum difference with the hillslope bedrock

CSPDF around 8 Ma (Fig. 8b). Therefore, constraining the hillslope sources for the bedrock SPDF allows us to better identify storage locations of sediment in comparison to the catchment bedrock SPDF (Fig. 8a-b).

The mean AFT detrital SPDF (Fig. S8a) shows a similar tendency of over-representing old ages (> 25 Ma) and under-representing mid-range ages (~16 Ma) compared to the hillslope bedrock SPDF. However, the mean AFT detrital SPDF differs

from the mean detrital AHe SPDF as it does not show a plateau-like form. Indeed, two age peak components occur at ~16 Ma and at ~40 Ma. However, the detrital cumulative distribution (Fig. S8b) is also shifted toward older ages, with a maximum difference at 24.5 Ma. The difference between the AHe and AFT age distributions is attributed to the different age-elevation

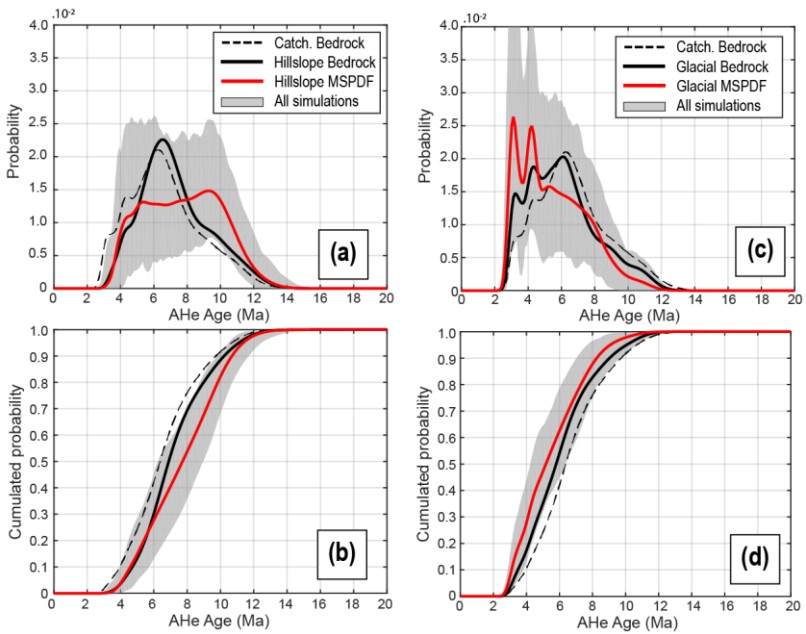

**Figure 8.** Detrital AHe age distributions of the frontal moraine, and their cumulatives, for models considering only ice-free hillslope sources (a-b) and glacial sources (c-d). The grey shaded area represents the range of the inferred 10,000 detrital SPDFs from the sampling process. MSPDF: Mean Synoptic Probability Distribution.

relationships with less spatial age resolution for the AFT system (Fig. 4a). Overall, the hillslope SPDFs reveal that the relevant source region ages tend to have older ages for both AHe and AFT, which is in accordance with the hillslope hypsometric

distribution (Fig. 4b).

We now consider particles originated from the glacial sources (Fig. 7a), which we define to be all the ice-covered areas. The associated detrital AHe ages range now from 2.7 to 12.1 Ma, according to the hypsometric distribution of glacial source part of the age-elevation profile (Fig. 4), spanning the first ~85% of the total age range of the catchment. The maximum age reflects the maximum elevation reached by the ice (i.e. ~3420 m). As before, we calculate a new bedrock SPDF based on the glacial

source distribution (Fig. 8). The detrital SPDFs, representing particles of glacial origin in the frontal moraine, show a maximum range of inferred SPDFs at ~3.3 Ma (Fig. 8c). We relate this range to the constant 10% relative age uncertainty applied, as the range of inferred detrital SPDFs tend to increase with younger ages and younger ages have smaller absolute uncertainties.

Compared to the glacial bedrock SPDF, the mean glacial detrital SPDF over-represents young ages (< 5 Ma), in contrast to the hillslope source as we might expect. Next, we observe a relative under-representation of ages ~6.3 Ma and for ages older than 8 Ma, consistent with a storage of sediments at mid-elevation. This is reflected in the cumulative detrital distribution (Fig. 8d), which is shifted towards younger ages and does not intersect the glacier bedrock cumulative distribution. The maximum difference between the two cumulative distributions is at 4.6 Ma.

The detrital AFT age distribution (Fig. S8) show the same tendency in over-representing young ages (age peak ~16.5 Ma) compared to the glacier bedrock distribution and by under-representing old ages (>28 Ma). The detrital CSPDF is also shifted toward younger ages, where the maximum difference with the bedrock CSPDF is observed ~28 Ma. We notice however smaller differences between the detrital and the bedrock distributions for the AFT than for the AHe system. Overall, the results show that older ages mainly reflect hillslope sources and younger ages, the glacial sources, which is expected given the age-elevation profile and hypsometric curves of the two sources (Fig. 4b). However, they also show that estimation of the bedrock distributions of the sources helps better constrain the role of the ice transport, and also better identify the predominant source in the sediment at the sampling site.

## 3.5 Detrital signature using a landscape evolution model with non-uniform erosion

To illustrate how the bedrock distributions of the sediment sources can help identify the dominant source signal at the sampling site, we consider now a model with non-uniform erosion. We use erosion laws presented in Sect. 2.1 with parameters listed in Table 1, which leads to the erosion pattern in Figure 9a. Eroded particles are thus a mixture of hillslope- and glacier-origin sources. The parameters of the erosion laws were calibrated by a trial-error process to obtain a mean erosion rate over both source regions of 1 mm yr$^{-1}$ for a more consistent comparison with the constant erosion rate models. Obviously, variations occur locally about this mean value (e.g. maximum erosion rate is ~31 mm yr$^{-1}$, Fig. 9a) and the calculated standard deviations for the mean erosion rates for the hillslope and glacier sources are $1 \pm 0.29$ and $1 \pm 2.9$ mm yr$^{-1}$ respectively. Variations in ice erosion result mostly from the distribution of effective pressure that controls the opening of cavities (Eq. 3 and Eq. 6) and therefore from the length on which the basal shear stress is applied (Eq. 1 and Fig. S1). Consequently, there are some ice-covered areas that do not erode at all. The range of detrital ages for the glacier-origin particles spans 9 Ma from 2.7 to 11.7 Ma, whereas those for the hillslope-origin particles range from 3.5 to 13.8 Ma.

The mean detrital SPDF shows a large age peak ~5 Ma, and a smaller one ~10 Ma. The glacial bedrock distribution predominates for ages <5.5 Ma (Fig. 9c), suggesting a glacial source for the first age peak. This is confirmed by the high erosion rates located where the AHe ages are around 5 Ma (see Fig. 9a-b and Fig. S6a). The second age peaks (10 Ma) best match with the hillslope bedrock distribution, which would suggest a dominant hillslope source contribution. This assumption is also confirmed by the erosion rates ~1.80 mm yr$^{-1}$ where AHe ages are ~10 Ma (Fig. 9a and Fig. S6). The mean detrital SPDF also shows a low at ~7-8 Ma that we attribute to both the effect of storage at mid elevation (Fig. 6b) and the erosion pattern, as the relevant elevations are represented by small tributary glaciers with low erosion rates (<0.07 mm yr$^{-1}$). A similar comparison can be done with the CSPDFs (Fig. 9d), although the curves are smoother. Thus, taking into account the bedrock

age distributions of the different sources increases the potential to discriminate these sources in a detrital sample. However, we stress that the bedrock distributions are based on the assumption of spatially uniform erosion across the catchment. Non-uniform erosion leads to different bedrock SPDF for the sources, and also different expected sediments source distributions.

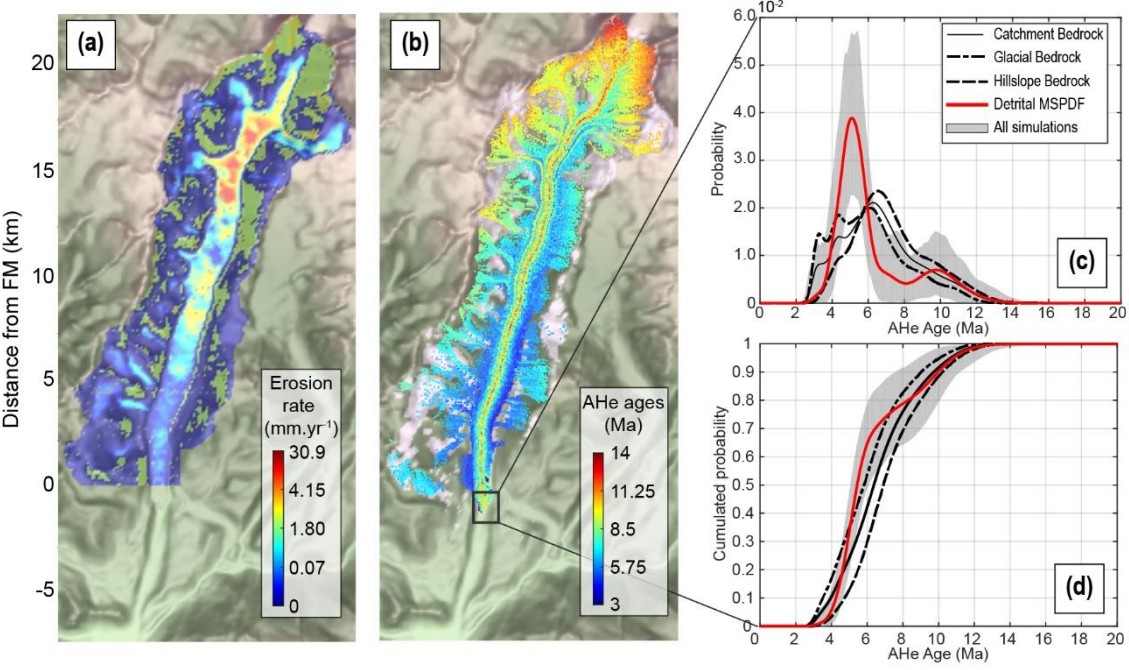

**Figure 9:** Non-uniform erosion model experiment, with (a) the erosion rate, (b) the spatial AHe age distribution of particles, (c) the detrital AHe age probability distributions and (d) their cumulative for the frontal moraine. The grey shaded area represents the range of the inferred 10,000 detrital SPDFs from the sampling process. The black square in (b) shows the frontal moraine (FM) location. MSPDF: Mean Synoptic Probability Distribution Function.

5   The mean detrital AFT SPDF (Fig. S9) is in agreement with a glacial source of particles for AFT ages ~16 Ma, where the erosion rates are highest (Fig. S9a), but this is much less pronounced than for the AHe system. Moreover, it is difficult to make distinction between the two sources (hillslope vs glacier) for ages >30 Ma, as the mean detrital SPDF closely match all the bedrock SPDFs. We relate this pattern to the large relative uncertainties (i.e. >30%) we considered for the AFT ages, that over-smooth the SPDF, especially for older ages. The mean detrital CSPDF shows no significant differences with the catchment
10  bedrock CSPDF and suggests uniform erosion across the catchment, which contrast to the conclusion inferred for the AHe system, i.e. a major contribution of the glacier source. This difference, due to the different age-elevation relationships and age uncertainties between the two thermochronometers, reduces the ability to track the source of particles with AFT data in this example.

## 4 Discussion

### 4.1 Limitations of the particles transport model

It is important to note that the sediment particles in our model are passively transported and do not interact with the ice. For instance, the concentration of debris in the basal ice can influence the rheology of the ice and the friction of the bed, which both impact the ice flow (e.g. Hallet, 1981; Iverson et al., 2003; Cohen et al., 2005). Surface debris-cover can also shield the ice from solar energy and, in turn, reduces the ablation rates and increases the local mass balance of the glacier (e.g. Östrem, 1959; Kayastha et al., 2000; Rowan et al., 2015). As iSOSIA does not perform a full 3D computation of ice deformation, the horizontal velocity of particles is parametrized as a simple polynomial function (i.e. Eq. 9) of the average ice velocity (i.e. for englacial particles). Sediments can move vertically only by the accumulation of new ice or by melting, whereas natural particles can also move due to internal ice deformation, including thrusting and folding (Hambrey et al., 1999; Hambrey and Lawson, 2000). Surface debris may also fall in crevasses and therefore be incorporated in the basal ice (Hambrey et al., 1999). Additionally, our models omit debris comminution during transport. This process may have important implications for sampling strategies as block sized sediments formed by, e.g. quarrying, potentially originate close to the sampling site (i.e. have a short travel distance), but often are not sampled. This could bias detrital SPDFs, that are only representative of the sampled size fraction of sediments (mostly sand size) and thus sources located high in the catchment (i.e. larger travel distances). Lastly, sediment transport by rivers and sub-glacial drainage is neglected in the models presented here. However, it has been shown that a large part of basal debris can be evacuated by meltwater, on a seasonal timescale (Collins, 1996; Kirkbride, 2002; Swift et al., 2005; Delaney et al., 2018). Not allowing for this factor potentially impacts the transfer time of our sediment particles to the glacier front and we return to this point below. Additionally, water mixes sediments efficiently and may reduce the role of advective sediment transport in the ice. The role of sediment mixing by water has been investigated by Enkelmann et al. (2015) and they conclude that sampling the pro-glacial river (Ehlers et al., 2015) is similar to sampling through the ice-core terminal moraine, meaning that sediments in glacial outwash have a greater potential to be mixed by meltwaters. We emphasise that our simple particle transport model is a starting point for studying the behaviour of sediment transport by ice, and integration of other processes of transport should be considered in future studies.

### 4.2 Effect of sediment transport by ice

Despite the limitations mentioned above, some first-order insights related to ice dynamics can be identified. Firstly, our ice transport model for the Tiedemann glacier implies an equilibrium time in terms of the relative proportions of particle sources in the frontal moraine of the order of $10^3$ years ($\sim$1500 years). This timescale is of the order of the characteristic timescale developed by previous studies (Johannesson et al., 1989; Oerlemans, 2001; Roe and 0'Neal, 2009; Herman et al., 2018) and mainly reflects the glacier dynamics. Incorporating transport of sediments by meltwater, this timescale would probably reduce for glaciers with well-developed drainage systems, as large part of glacier basal debris seems to be evacuated by the subglacial drainage system (Collins, 1996; Kirkbride, 2002; Swift et al., 2005; Delaney et al., 2018). However, the types of debris forming

frontal moraines vary from glacier to glacier. For instance, debris-covered glaciers are dominated by supraglacial debris that are mainly transported passively by ice (Benn and Evans, 2013). Terminal moraines of debris-covered glaciers are thus mainly built by a dumping process of debris from the ice surface, and thus reflect glacier dynamics, especially if the glacier remains stable for a long period of time (Sharp, 1984; Lukas, 2005; Hambrey et al., 2008; Benn and Evans, 2013). Moreover, some

sedimentological studies on terminal and lateral moraines have shown limited amounts of glaciofluvial-related facies (e.g. Winkler and Matthews, 2010; Bowman et al., 2018; Ewertowski and Tomczyk, 2020), as it is the case for the Tiedemann glacier (Menounos et al. 2013). Therefore, the timescale for building the frontal moraine would strongly depend on (i) the availability of subglacial debris vs supraglacial debris, and (ii) the glacier dynamics. We recommend that future studies should focus on the timescales for building the terminal moraine, but we also highlight that, in some glacial areas, this timescale could

impact the interpretation of detrital SPDFs from such glacial features, e.g. by reflecting a transient phase of sediment arrival (as shown in Fig. 6).

Secondly, zones of slow ice flow, such as in small tributary glaciers, may act as sediment traps as shown by particles remaining in elevated areas after 8500 years of simulation. These zones of slow ice motion occur mostly around the ELA and are associated to low values of local topographic slope. This is consistent with the suggestion that glacial erosion is more efficient

around the ELA (e.g. Egholm et al., 2009; Brozović et al., 1997; Anderson et al., 2006; Steer et al., 2012), which leads to limit local relief and slope around the ELA and in turn to limit ice sliding velocity. This also highlights morphodynamic feedbacks controlling, here by a trapping effect, the detrital distribution of thermochronological ages. Therefore, only ~44% of the total initial number of particles reach the frontal moraine after 8500 years of transport (i.e. the maximum time of our simulation). About 25 % of the particles are stored in the lateral moraine (Fig. 5d) and the remaining ~31% are trapped at higher elevations,

and therefore have residence times greater than 8500 years. The robustness of the results presented so far to different parameters has been tested by varying the glacier size, and the hillslope diffusivity for particle transport. The results are presented in Sect. 2 of the supplementary materials. Overall, the conclusions are similar to those already discussed, and show equilibrium times for the frontal moraine of the same order, and around 53-60 % of the total sediments stored higher in the catchment

To illustrate the effect of low velocities on transport times in more detail, we compute the average transfer time as a function of source location for particles that reach the frontal moraine (Fig. 10). The results show that the time required for a particle to reach the frontal moraine is not simply proportional to the distance with the source location. Indeed, some particles formed near the frontal moraine may take more than 3000 years to reach the glacier front due to velocities close to zero (Fig. 10). In contrast, some particles formed far from the frontal moraine have transfer times less than 1000 years due to averaged-velocities

greater than 15 m yr$^{-1}$ (Fig. 10b). The proximity of the source to ice streams with high velocities explains this pattern of transfer times (Fig. 3c). The average transfer times for particles formed along hillslopes or glaciers are 1825.4 $\pm$ 1914.7 and 1084.9 $\pm$ 1014 yrs, respectively, but show high variability. This difference is controlled i) by the longer average distance of the hillslopes to the frontal moraine, 15.26 $\pm$ 7.19 km, compared to glaciers, 13.85 $\pm$ 6.15 km, and ii) by spatial variation in ice flow velocities and storage of sediments in small tributary glaciers. Overall, sediment transfer times of our models are strongly

influenced by the spatial distribution of small tributary glaciers, implying an important control of glacier sizes on the delivery of sediments to the main glacier transport system.

In the case of uniform erosion and a simple relationship between thermochronological age and elevation (Fig. 4a), we expect the detrital thermochronological signal associated to each source to mimic its associated hypsometric distribution. However, the resulting detrital SPDFs differ from that expected from the hypsometric distributions (Fig. 7). The lack of the ages in the detrital SPDF corresponding to the peak observed ~1500-2000 m in the hypsometric curve, may be explained by (i) the storage of a major part of the sediments (i.e. 56%) outside of the sampling site, and (ii) by the patterns of transfer times for particles that reached the frontal moraine (Fig. 10). The generality of this conclusion is limited by our sediment transport model, as mentioned earlier, and the discrepancy observed between the detrital SPDFs and the expected distribution (i.e. bedrock SPDF) can be reduced for glacier with well-developed subglacial drainage system.

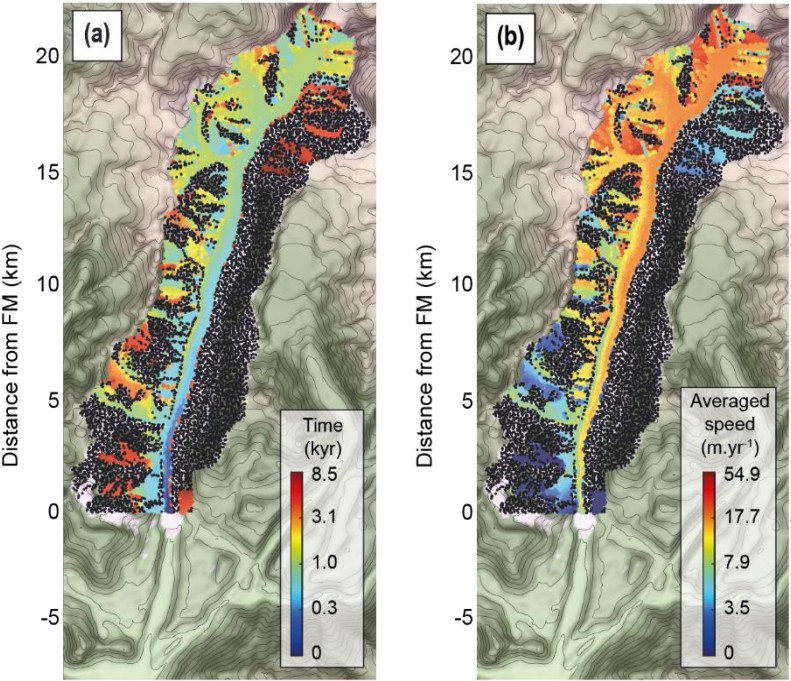

**Figure 10:** Characteristics of particles transfer to the frontal moraine, (a) the time needed for each particle to reach the frontal moraine; (b) their averaged speed. Black dots are particles that did not reach the frontal moraine (FM).

### 4.3 The role of detrital sources: glaciers or hillslopes

According to our model for the Tiedemann glacier, each type of detrital source, i.e. hillslope or glacier, displays a different average hypsometry (Fig. 4b). Glaciers mostly represent valley floor while hillslopes mostly represent elevations greater than ~1800 m. Hillslope sources contribute to older ages in the detrital SPDFs, as expected from the age-elevation profiles and the

range of elevation of hillslope source (~700-4000 m). The glacial detrital SPDF (Fig. 8) reflects mainly the younger ages from the valley floor, and a lack of older ages in the elevation range occupied by glaciers (~530-3420 m).

When estimating glacial erosion rates from sediment flux measurements it is important to distinguish glacier-origin debris from supraglacial debris. Previous studies used cosmogenic nuclides concentrations (e.g. Guillon et al., 2015) or U-Pb ages

(e.g. Godon et al., 2013) on sediments to discriminate between sources. Here, we tested a simple approach by characterizing the bedrock age distribution of different sources and compared them with the detrital age distribution. The model considering non-uniform erosion gave promising results for identifying contribution of different sediment sources to the frontal moraine (Fig. 9).

Despite identical averaged erosion rates for the two sources (i.e. 1 mm yr$^{-1}$), the resulting shape of the mean detrital SPDF is

the result of higher local glacial erosion rates compared to the more diffusive erosion on hillslopes. Therefore, the glacial source locally produces a high amount of sediment particles, with similar thermochronological ages, which are ultimately transported to the frontal moraine. This explains the high peak observed ~5 Ma (Fig. 9) and the lower peak ~10 Ma corresponding to hillslope sources. Furthermore, the proximity to the depositional site of the sources with high glacier velocities also contributes to the magnitude of the age peak components observed in the detrital SPDF (Fig. 10). Therefore,

detrital SPDFs of glacial features such as the frontal moraine are likely to over-represent glacial sources compared to hillslopes sources if driven by locally high erosion rates. Finally, a large part of sediment particles produced on hillslopes originates from around 2800 m as illustrated by the older age peak component at ~10 Ma (Fig. 9c).

## 4.4 Implications for detrital sampling strategies

A sampling strategy equivalent to the one considered in this study (i.e. randomly sampling the entire frontal moraine) has the

potential to capture more bedrock age components, as proposed in previous studies (e.g. Enkelmann et and Ehlers, 2015). To illustrate this effect, we now consider different sampling strategies. We perform an additional experiment, with uniform erosion, where sampling occurs in four different regions of the frontal moraine (Fig. 11). The process of sampling is the same as for previous models, i.e. we randomly collect 105 particles within each region, produce a SPDF and repeat this process 10,000 times to infer a mean detrital SPDF. The resulting mean AHe detrital SPDFs and CSPDFs (Fig. 11) show significant

variability. Sampling region one mostly captures young ages (<6 Ma) and therefore the glacier source, while hillslope sources, with older ages (>6 Ma), are under-represented. Sampling region two, located closer to centre of the moraine, includes an older age component (>6 Ma), which leads to a better fit with the mean detrital SPDF obtained by sampling the entire frontal moraine. Statistical tests performed on the CSPDFs confirm this similarity (see Table 1). However, the different sensitivity of these statistical tests complicates interpretation of these results. Finally, the regions three and four show similar distributions with

an over-representation of young (<4 Ma) and old ages (>7 Ma), and a gap in mid-range ages (between 5 and 7 Ma) representing intermediate elevations (1500-2000 m). Old ages (red dots in Fig. 11a-b) are mostly transported in the central part of the main glacier and deposited in the central part of the frontal moraine (regions two, three and four). An exception occurs with regions three and four due to deviation of ice streamlines (Fig. 11b). This deviation is also responsible for significant deposition of

sediments in the lateral moraine upstream. The trend of decreasing young age component in the detrital SPDF across the frontal moraine is seen for the AFT system, although the differences are less pronounced. However, this trend seems to be supported

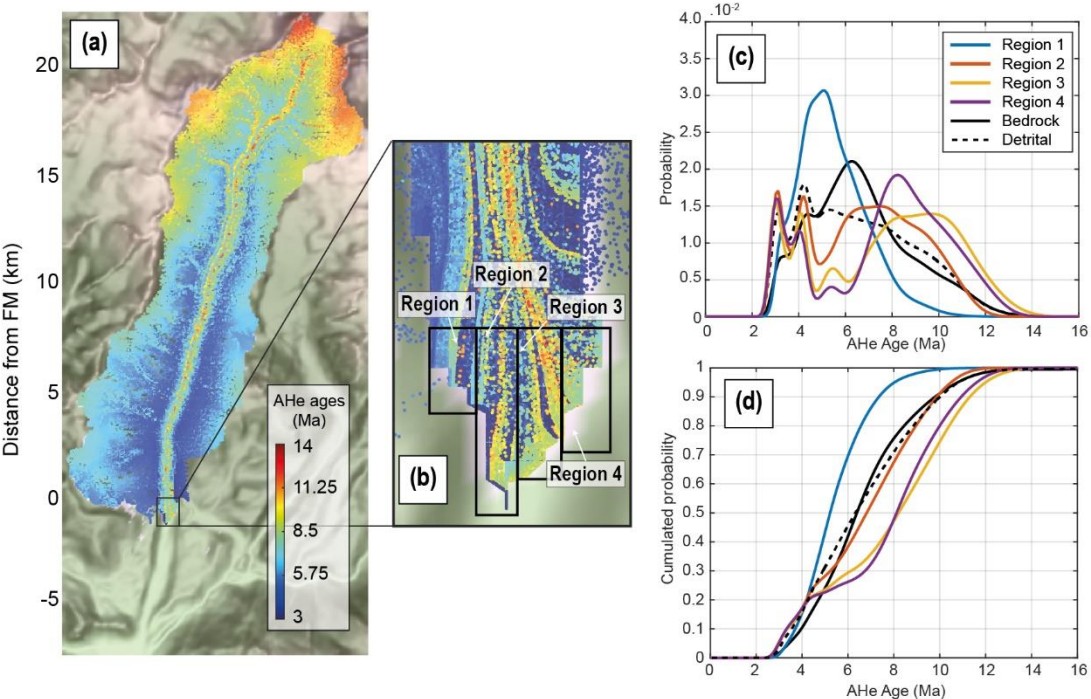

**Figure 11:** Detrital AHe age distributions of the five regions seen in (b, black rectangles) from the experiment considering uniform source of particles. Spatial distribution of particles with their AHe ages associated (a), with a zoom to the frontal moraine area in (b). The density plots and their cumulative distributions are shown in (c) and (d). The dashed black line is the mean detrital SPDF for the frontal moraine (FM).

by a comparison with true ice-cored terminal moraine AFT data from Enkelmann and Ehlers (2015) (see A2). We observe that, from left to right, the detrital age distributions within the frontal moraine incorporate older ages. However, we are aware that the method used to build SPDFs with a Gaussian kernel (Eq. 11) tends to break down with high (>30%) relative uncertainties (Brandon, 1996), as for the presented AFT data. We use this method for simplicity and to facilitate comparison with the synthetic AFT SPDFs. However, the original data also show the same tendency of age components getting older across the terminal moraine, and statistical tests applied on original data (Enkelmann and Ehlers, 2015) support the high variability of the detrital AFT age distributions across the terminal moraine regions (see Table A1). Overall, local sampling within the frontal moraine or upstream along a transverse transect leads to a higher variability in the inferred detrital age distributions. Even if some regions may show better agreements with the bedrock SPDF (region two), randomly sampling small patches of sediments through the moraine captures most of the age components on average and seems a better strategy overall.

## 4.5 The effect of age uncertainties and age-elevation profiles: comparing AHe and AFT

For all of our models, we concluded that detrital age distributions resulting from the AFT ages were less informative than those from the AHe ages. The differences between the two systems occur for two main reasons. First, the age-elevation profiles differ. For the AFT profile, the youngest ages are not at the lowest elevation but occur at ~2000 m of elevation. The slope of the AFT age-elevation relationship is almost vertical (and actually negative) for elevations lower than 2000 m. This low age-elevation gradient leads to a reduction in source identifiability, i.e. similar ages can come from a large range of elevations. Secondly, the high relative age uncertainties (i.e. >30%) in the AFT data smooths the SPDF and decreases the resolvability of age components in the age distribution. Consequently, these two characteristics can make the AFT system perhaps less useful for tracking erosion patterns, as seen for the case of non-uniform erosion. However, in some cases the AFT distribution can still capture first-order behaviour of sources if combined with bedrock age and age-elevation distributions, depending on the distribution of erosion across the catchment.

## 5 Conclusions

In this study, we have presented a numerical approach to investigate the effect of sediment sources (hillslopes vs glaciers), ice transport and deposition on the distribution of thermochronological ages found in a frontal moraine. We applied this approach to a glaciated catchment which presents simple morphological characteristics: the Tiedemann glacier (British Columbia, Canada) in steady-state. Firstly, the presence of small tributary glaciers with very low velocities, may act as traps of sediments and delay particle transfer to the glacier front. These low velocities may result from a morphodynamic feedback between the location of the ELA and the bed slope in direction of sliding. Secondly, the transfer times of sediments are influenced by the proximity of their sources to the ice streams showing high velocities. Indeed, sediments located in elevated areas may experience lower transfer times than sediments produced closer to the sampling site. Thirdly, horizontal separation of ice flow lines can produce lateral moraines that may store a significant amount of sediment produced upstream. This implies that frontal moraines may include sediments that contain thermochronological signatures from limited parts of the total catchment. To address this problem, lateral moraines of the same age deposition as the frontal moraine can be also targeted for complementary sampling, therefore incorporating more age components.

Sediment transport by ice can lead to differences in the detrital age distributions compared to the bedrock age distribution, for instance by undersampling mid-altitude age components. This could lead to misinterpretation of regional erosion patterns. Moreover, strategies considering local sampling of sediment in the frontal moraine show variable detrital age distributions, that predominantly reflect the variability of local sources upstream. In principle, this may allow us to directly associate particle sinks, e.g. moraines, to their sources. In contrast, randomly sampling through the frontal moraine potentially captures more age components, providing a more representative picture of the whole catchment. Therefore, we suggest the sampling strategy should be designed according to the question being addressed. Furthermore, we systematically compared two thermochronological systems, AHe and AFT, with different but coherent age-elevation profiles and different relative age

uncertainties. While the first factor plays a role in the ability to track sediment sources, the second factor impacts on the precision of SPDFs. However, characterization of the spatial variability of source contribution between different sampling sites may still be possible depending on the distribution of erosion.

Overall, our numerical approach offers novel insights on the application of detrital thermochronology to glaciated catchments and on the role of long-term exhumation, modern erosion, and past sediment transport. However, as we have stated earlier, this study considers simple laws for ice motion and sediment entrainment and neglects sediment transport by the glaciofluvial system. Clearly, directions for future studies are the role of subglacial hydrology and, plastic deformation of ice on sediment transport and on the role of meltwater in the building of terminal moraine. Finally, we have only considered one single alpine catchment, our results may not be applicable to catchments showing more complexity (e.g. tributary glacier valleys).

## Author contribution

Maxime Bernard developed the model code for the computation of thermochronological ages and SPDFs, designed the experiments, and prepared the manuscript with contributions from Philippe Steer and Kerry Gallagher. David L. Egholm provided the model iSOSIA with the associated knowledges to allow its use. He also contributed to the final version of the original draft.

## Competing interest

The authors declare that they have no conflict of interest.

## Acknowledgements

We thank Eva Enkelmann for having shared the AFT data from the Tiedemann glacier, which have been used to produce synthetics AFT ages. We also want to particularly thank the Insitut Français du Danemark (IFD) for having helped exchanges between Rennes and Aarhus with their financial support. Finally, we address special thanks to Peter Van der Beek, Stephane Bonnet, Benjamin Guillaume, and Pierre Valla, for their advices and guidance that ultimately conducted to this study.

## Appendices

**Table A1:** Statistical tests results from Kuiper and Kolmogorov-Smirnov (KS) tests, with associated p-value for the modelled detrital CSPDFs shown in section 4.4. Each modelled detrital CSPDF (Region 1-4, and frontal moraine) is tested against the modelled catchment bedrock CSPDF. Frontal Moraine corresponds to the mean detrital CSPDF of the entire moraine. The variability of detrital SPDFs of the original data from Enkelmann and Ehlers (2015), p-values for S10-S14, has been tested by comparing the original ice-cored detrital SPDFs against the detrital age distribution of the glacial outwash sample (9TETG15) presented in Ehlers et al. (2015). Highlighted in black are contradictions or p-value close to the alpha level (0.05) between the two statistical tests about the similarity between the corresponding

detrital CSPDF and the catchment bedrock CSPDF (or glacial outwash in case of S10-S14). 1: the two distributions are different, 0: the two distributions are similar.

| | AHe | | AFT | |
|---|---|---|---|---|
| | Kuiper test/p-value | KS-test/p-value | Kuiper test/p-value | KS-test/p-value |
| Region 1 | 1 / 1.16x10$^{-5}$ | 1 / 4.10x10$^{-7}$ | 1 / 1.37x10$^{-6}$ | 1 / 3.25.10$^{-8}$ |
| Region 2 | **0 / 0.057** | **0 / 0.26** | **0 / 0.09** | 1 / **0,01** |
| Region 3 | 1 / 1.63x10$^{-9}$ | 1 / 4.80x10$^{-8}$ | **1** / 6.42x10$^{-22}$ | 1 / 3.26x10$^{-24}$ |
| Region 4 | 1 / 4.46x10$^{-11}$ | **1** / 4.68x10$^{-9}$ | 1 / 5.78x10$^{-20}$ | 1 / 3.38x10$^{-22}$ |
| Frontal Moraine | **0 / 0.50** | **0 / 0.55** | 0 / 0,87 | 0 / 0,31 |
| S10 | - | - | 1 / 8.83x10$^{-16}$ | 1 / 1.30x10$^{-17}$ |
| S11 | - | - | 1 / 4.99x10$^{-9}$ | 1 / 4.03x10$^{-7}$ |
| S12 | - | - | **1 / 0.03** | **0 / 0.30** |
| S13 | - | - | 0 / 0.25 | 0 / 0.38 |
| S14 | - | - | 0 / 0.52 | 0 / 0.10 |

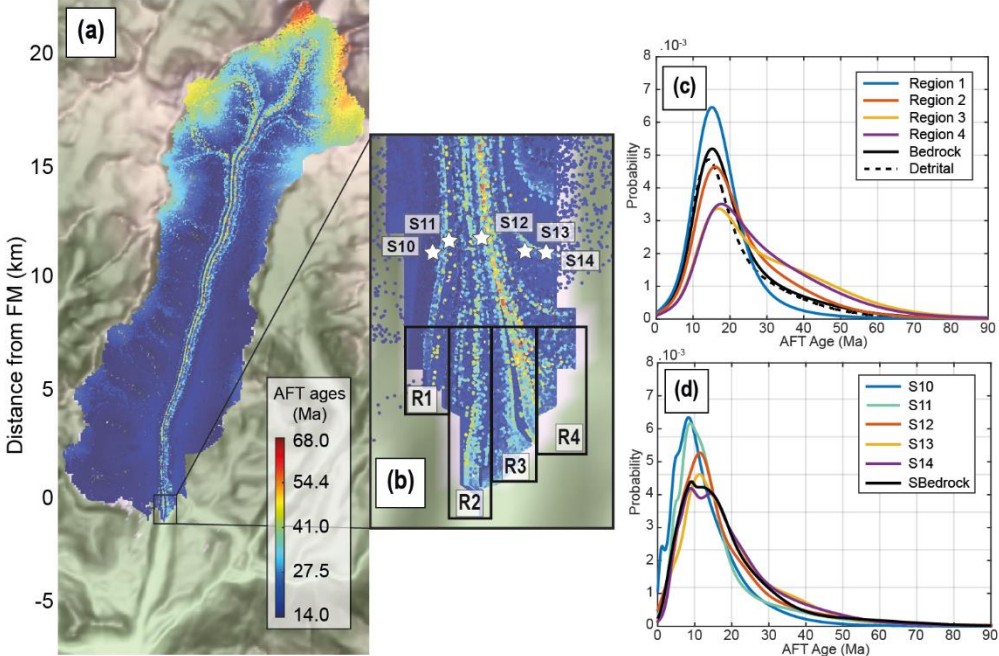

**Figure A2:** Detrital AFT age distributions of the four regions seen in (b, black rectangles) from the experiment considering uniform source of particles (a). The density plots of modelled detrital AFT ages are shown in (c). The dashed black line is the mean detrital SPDF for the frontal moraine (FM). The ice-cored detrital SPDFs from Enkelmann and Ehlers (2015), is shown in (d). Each sample (S10-14) is located in (b) by the white stars, and the age distributions have been built using the method explained in Sect. 2.4. The bedrock SPDF in (d) results from the bedrock single grain age distribution presented in Enkelmann and Ehlers (2015).

**Code availability**

The iSOSIA version (iSOSIA_3.4.7b) and the external routine used to compute the detrital age distributions are publicly available here: https://github.com/davidlundbek/iSOSIA.

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
