# Peer review of "Modelling the effects of ice transport and sediment sources on the form of detrital thermochronological age probability distributions from glacial settings"

_Earth Surface Dynamics, 2020_

## Referee Comment (RC1) · Todd A. Ehlers (Referee) · 16 Mar 2020

SUMMARY:

This is a very interesting study that advances our knowledge of how glacial (and to a lesser degree hill slope) processes can be evaluated using detrital thermochronometer samples from end moraines. The authors present results from a well-developed 2nd order glacier flow and erosion model. The model is roughly tuned to simulate the Waddington Region (Tiedemann Glacier), British Columbia where a large data set of

bedrock, ice cored moraine detritus, and glacial outwash detritus, apatite (U-Th)/He and apatite fission track ages have been published (i.e. data from Ehlers et al. 2015; Enkelmann and Ehlers, 2015). I collected the samples the authors use in this study and am very familiar with the study area (and data). The authors use particle tracking in the model to determine what the age-distribution of eroded material will be in the end moraine. The primary results of the study focus on the time scale required to reach an equilibrium age distribution in the frontal moraine, and also spatial variations in upstream catchment erosion.

The content of this manuscript is highly relevant for this journal, and the results are new and significantly advance our knowledge of this topic. The topic addressed is important because it covers the exciting field of linking detrital thermochronology to geomorphic processes. Overall, I am very supportive of eventual publication of this manuscript. However, in it's current form, there are many aspects of the study that are difficult to understand, and some additional analysis and potentially a few model sensitivity simulations are required before it should be considered for publication. I've provided below detailed comments on areas of the manuscript where I struggled to understand the results and interpretations. In many cases, my suggestions may not be correct -but they are based on my understanding of the text and nevertheless highlight portions of the manuscript that need additional clarification for all readers. The authors should feel free to contact me if any of the comments are not clear.

Based on the above, and following, comments, I recommend major revisions and re-review.

GENERAL COMMENTS:

G1. The general validity (beyond the simulations presented) of the authors results and interpretations is difficult to assess. If I understand the text and Table 1 correctly, the authors present a set of simulations with with only one set of parameters. These results are nicely presented and described, but the primary interpretations of the paper (e.g.

1,500 yr equilibrium time for observed age-distributions; sediment trapping in tributary glaciers, hill slope vs. Glacial contributions to observed age-distributions) are based on this single (?) set of chosen parameters. The results would be generalisable and more broadly applicable if a small set of additional simulations with a sensitivity analysis were included. For example, picking the least well constrained climate, ice, and hill slope parameters and picking reasonable values above and below what is shown in Table 1 should be considered to evaluate how robust the interpretations are in the text. I suggest these additional figures be shown in the supplementary material and then referred to in the main text, or highlighted in a new discussion section with a new figure that compare results.

G2. The results of Enkelmann and Ehlers (2015, Chem Geo) compared AFT ages across the ice cored moraine, outwash, and older moraines in front of the modern glacial terminus. The ice cored moraine and outwash samples were statistical identical (Fig. 6E), meaning that multiple ice cored samples when combined produce the same grain-age distribution as outwash. "slight differences" were observed between the individual ice cored moraine material in this study (Fig. 4 of Enkelmann and Ehlers). While the authors of this study (Bernard et al) explicitly say they are not trying to match observations in their study, in the discussion section they end making statements that do compare / evaluate the observations. There are suggested revisions related to this:

G2-a. In the start of the paper when describing the previous observational studies in the area (and the bedrock data you use), please add text that makes it clearer how these data were collected and which data sets are or are not relevant to how the model is setup and why. For example, the Enkelmann and Ehlers 2015 data are well suited for comparison to the model results (they come from ice cored moraine material at the end of the glacier). In contrast - the Ehlers et al. 2015 data are from glacial outwash and are not suitable for comparison to how the model is setup. The modeling approach does not track particles through the subglacial hydrologic system. No comparisons to this later study should be made in this manuscript.

G2-b. The concluding discussion section (4.4) and Fig. 10 make model predictions that are more or less comparable to the observations of Enkelmann and Ehlers, 2015. Although some qualitative comparisons are made in the text, the manuscript would be much stronger if the data from Enkelmann and Ehlers, 2015 were also plotted in Fig. 10c and similarities / discrepancies are discussed.

G3. The model setup and description needs to be clearer. It is not clear how erosion is done for hill slopes and how ages from hill slopes are mixed with the glacial sourced ages. It would greatly improve a readers understanding of this study if the coupling (and particle tracking) between hill slopes and glaciers was better explained, and also the sensitivity of their results (e.g. Fig. 7) to some of these parameters (e.g. see also comment G1 above). I also don't understand exactly how the 'uniform' erosion model was calculated with the ice model (see detailed comment below). The methods sections needs to explain the model setup for all this better. Section 2.1 (end) explains a non-linear diffusion model is used for hill slopes, but it's not described in enough detail how the sediment transport is done in this part of the landscape. The paper later on nicely explores how hill slope vs. Glacial sediment sources impact detrital cooling ages. Thus, some additional text in the methods section would make it much easier to understand the model results. As it's written now, I could not reproduce your results if I wanted to.

G4. Comparison of distributions should include statistical tests. Section 4.4 (see also detailed comment 26 below) presents an analysis of how representative point samples across a moraine compare to the mean of all samples and the bedrock distribution. This section is very useful for understanding how and where people could sample. However, the analysis does not statistically evaluate if the different synthetically sampled distributions are the same or not. Visual / qualitative comparisons of distributions is dangerous, and I suspect (based on experience) that several of the distributions show in Fig. 10c will be statistically identical. If this section remains in the paper (which I hope it does), it's important the authors conduct a simple KS or Kuiper test

to see if in fact the different distributions show in figure 10c are different or not. This comment could also potentially impact the conclusions presented in section 4.5. The text in section 4.4 and 4.5 is good, but simply needs support from a more quantitative comparison. Finally, as mentioned above for comment G2, this model result should be compared to observations that were collected from roughly this same area (Enkelmann and Ehlers, 2015).

G5. The text is in general well written and clear. However, there are many small wording issues / grammatical problems throughout the text (e.g. missing articles, subject/verb agreement) that are understandably hard to catch for a non-native speaker. I recommend a native speaker/co-author give the text another thorough read to correct these.

SPECIFIC COMMENTS:

0. For clarity - all axis labels with 'Age' on them should say what age (e.g. "AFT Age") you are plotting since you work with two different systems in this study.

1. Abstract should make it clear if the langragian particle tracking is only for ice flow, or subglacial water.

2. Page 2 paragraph starting at line 17: Also relevant to the content of this paragraph, and the study area investigated is the study by Yanites and Ehlers 2016 that documents how glacial sliding relates to bedrock thermochronometer ages (in a neighbouring valley in the Coast Mountains). Yanites, B. J. and Ehlers, T. A.: Intermittent glacial sliding velocities explain variations in long-timescale denudation, Earth and Planetary Science Letters, 450, 52–61, doi:10.1016/j.epsl.2016.06.022, 2016.

Also - this manuscript is highly relevant to the following previous work and the authors should consider citing it in the introduction or discussion. Herman, F.,et al., 2018. The response time of glacial erosion. JGR - Earth Surface. https://doi.org/10.1002/2017JF004586

3. Page (pg) 3, line (ln) 12 - It might be worth clarifying here for readers that the Enkelmann and Ehlers 2015 studied sampled ICE across the ablation zone, and the Ehlers et al. 2015 study sampled glacial OUTWASH. This would help readers understand why you can not directly compare model predictions to one of these sets of observations..

4. Pg5, 6 model description. Hydrology effects on sliding are described here, but please also add a sentence or two that says if this is also included in the particle tracking for making SPDFs of cooling ages. Maybe you address this later.

5. Pg5, 6 - It is also important that this section says how you have calibrated the model for the subsurface hydrology. This aspect of the model is likely very important for how the data are interpreted, so some text on this aspect would be useful.

6. Pg.6 - Your approach assumes all sediment comes from quarrying. I'm more or less ok with this (note fine sediment fractions were present in what we sampled for this glacier). However, please explain here if you account for the comminution (breaking down) of plucked material. Why could this be important? If 20x20cm rock is plucked in the upper reaches of the catchment it will break down during transport and provide fine grain material that was sampled. If a 20x20 cm rock is plucked from 100 m from the sample point - it wouldn't show up in sample. The material sampled for the Tiedemann glacier ice cored moraine and outwash was a 'bulk' sediment sample, but with nothing greater than ∼2x2cm size in it. I would be great if your modeling approach accounted for comminution, but I'm guessing this is not the case. So this effect needs to be acknowledged and the potential implications of it discussed in a model caveats section.

7. Pg 6/7 - section 2.3. As indicated above, please explicitly state that water transport of detritus is not accounted for. Fluvial systems mix sediment very efficiently and the flow rates on these outlet rivers are high (rounded cobbles were in the river bed and appeared transported by it).

8. Pg. 8 section 2.3. Please add some text saying how the glacial mass balance (and climate inputs) are calculated and refer to your table.

**ESurfD**

Interactive
comment

9. Fig. 4c, d - I suggest labelling the x-axis with the age plotted (e.g. "AHe Age" for c). Also - for the caption, mention what uncertainty you used for making the PDFs since this influences the smoothness of the curves relative to panel B. Caption should also explicitly say the data come from glacial outwash, not moraine.

10. Pg12 ln7. Please say under what conditions the equilibrium state was calculated, and how closely it matches the present day thickness and length of the glacier.

11. Fig. 5. Please provide a more descriptive caption of what model this is at the start of the caption. Also - what are you actually doing with the 'hillslope' vs. 'glacial' parts of the catchment (Fig. 5a). It's not clear from the text (or caption) if you are also feeding hill slope material into the SPDFs calculated.

12. Pg13, ln1-2. Reword sentence please.

13. Fig. 6 - caption needs a starting sentence saying in general what is plotted and what model it comes from. Also- again for panel c, d - indicate the age type ("AHe Age") on the x-axis. Please do this for all other figures if this is also the case. It makes it much easier to read the figures quickly.

14. Pg.14 ln1. Please explain better what you mean by "model with uniform erosion". I'm confused because I don't understand how you made the ice model have uniform erosion - and where the uniform erosion was applied (e.g. Hillslopes and glacial areas?). Fig. 7e kinda gets at this, but the text should explain it better. Thanks.

15. Pg. 16 top. After reading this page (related to previous comment) I'm still confused - and some clarification is needed on this paragraph and what was actually calculated. The start of the paragraph needs to explain better what the objective of this comparison is (hillslope vs. glaciers) . Also ln3-4 are confusing because I thought this section only about uniform erosion, but this sentence says you are comparing a detrital SPDF to the uniform erosion model. Are you talking about the OBSERVED detrital SPDF? Perhaps make it clear throughout the text by always using 'observed' vs. 'modeled' detrital

SPDFs.

16. Pg16, ln10+. This paragraph is also unclear. After reading all of section 3.3 - I'm confused as a reader. Please rewrite this section to make it clearer of a) what is the logic behind the experiment / comparison conducted, b) what is the first and second order main trends in the results and what data / model results results (bedrock vs. Detrital you're looking at, and c) summary sentence(s) with the key observation to take away.

17. Maybe I missed it earlier, but after reading the results section - I think the methods section needs to be expanded some to explain how you look at (or calculate) hillsope vs. Glacial contributions to the detrital cooing ages.

18. Pg16 ln 29. I don't understand how "we computed a new...". How did you compute this? Assuming uniform erosion? Using a diffusion based hill slope transport law? Please elaborate.

19. Pg.17 ln5-10. This result is entirely dependent on the assume hill slope transport law used, and diffusivity,....right? Perhaps mention this, and also make it clearer (per my previous comments) how the curve in Fig. 7 is calculated.

20. Pg. 18 ln1. Several times in the paper you refer to or tune the model to an erosion rate 1 mm/yr. Why did you use this value? This should be explained in the methods section.

21. Pg.18 ln1-4. I don't understand this sentence and where these numbers you cite are coming from. For example, where is 31 mm/yr coming from? Where are the uncertainties in the numbers later in the sentence coming from?

22. Pg. 18. Ln6. Please rewrite this sentence. I don't understand it.

23. Pg. 19 ln1. "traducing"? Not sure what you're trying to say.

23. Pg 19 ln20. Please expand this thought (particle tracking is in ice, not sub-glacial

drainage). I would describe in general (qualitative) terms how the results could differ if subglacial outwash was sampled (e.g. how Ehlers et al. 2015 sampled). F

24. Pg. 19. Ln 24-25 "spatial erosion pattern can be biased on the detrital SPDF (e.g. Ehlers et al. 2015)". Please remove the "e.g. Ehlers et al. 2015". You may be correct that there is a bias, but you haven't shown this because the Ehlers et al. samples were from OUTWASH, not from ice. Also - your timescale arguments for bias later in the paragraph would likely be severely decreased if outwash is sampled because transit times of water to the outlet are significantly faster than for ice. So, please either show that Ehlers et al. 2015 have a bias in their outwash sample interpretation, or remove the reference to this paper to be fair. Final note concerning the general conclusions you are trying to make here about timescales - while your description is accurate for the simulations you present, it's hard to know if this is really a general result with any sensitivity tests to your model parameters presented in the study. Please consider adding sensitivity tests in the supplemental material and referring to them in the text.

Table 1 - please be more specific about what you mean by "variable" in some rows. I think what you mean is that these variables are 'internally calculated'

25. Pg. 21 ln29-30. This isn't really a surprise is it that moraines will more closely represent glacial erosion than hill slopes? Also - doesn't this statement and paragraph sort of contradict the previous section where you say there would be a bias towards low elevations in SPDFs? Maybe I'm missing something here - but perhaps more text (in the previous section about biases) relating to what section 4.3 is saying would help clarify things more.

26. Pg.22 section 4.4. This is nice that you done this analysis. However, to do this type of comparison you need to test if the distributions show in Fig. 10c are in fact statistically different. To do this, you need to apply a KS (or Kuiper) test of the distributions. Please include this type of analysis to see if these distributions really are in fact different at the 95% confidence level. This comment also potentially impacts the

interpretations presented in section 4.5.

27. Pg. 24 ln 1-2. This conclusion is correct for the simulations presented, but how variable is this result if there are variations in some of the model parameters. A sensitivity test of model inputs would help readers see if this result is general, or specific to your model simulations.

28. Pg. 24 ln22-23 "However, we also emphasis . . ." This statement should be removed or significantly expanded & justified. How many grains should be sampled is a significant topic on it's own, and this study doesn't address this. The typical "100" grains minimum that people site from Vermeesch is not actually correct for these types of detrital samples (we have a paper in preparation that goes through the statistics of number of detrital AHe samples needed for different catchment sizes and it's complicated).

29. Supplementary materials. There is useful material presented in the supplement, but the figure captions are a too sparse for some figures. Please expand the figure captions some to be more descriptive of what is shown. For example, what model simulation is shown in the figure, and perhaps also what section of the text the figure is relevant to.

Overall a nice piece of work. I enjoyed reading this study.

*** End of Review ***

---

## Referee Comment (RC2) · Anonymous Referee #2 · 15 Apr 2020

Here is my review of "The effects of ice and hillslope erosion and detrital transport on the form of detrital thermochronological age probability distributions from glacial settings" by Bernard et al. In this paper, a glacial-hillslope erosion model is used to predict distributions of detrital thermochronological data. The model is applied to the Tiedemann glacier in British Columbia, where detrital thermochronological data had been previously collected from glacial moraines and glacial outwash (Ehlers et al., 2015; Enkelmann and Ehlers, 2015). The authors conclude that 1500 years are required to reach an equilibrium for detrital particle age distributions. The modeling exercise is

interesting and the results nicely presented. However, the paper has one very important limitation that make the overall conclusions rather questionable, as I elaborate on below.

First and foremost, the model assumes that all the sediment transport happens within the ice – englacially or through sliding at the ice bedrock interface – using Eqn. 9 in the current paper. Unfortunately, it is often thought that the majority of sediments is transported out of the glacier through the subglacial hydrology system, which is not part of the model presented here. As explained by the authors, several processes govern sediment dynamics and bedrock erosion in the subglacial environment (e.g., Alley et al., 1997). Erosion obviously creates the sediments through quarrying and plucking as assumed and explained in the paper. Some of this sediment produced by glacial erosion can be accreted to the basal ice in locations such as overdeepenings and carried with the ice flow (e.g., Hambrey et al., 1199; Swift et al., 2018), but the majority of sediment produced by glacial erosion is transported by water flowing along the glacier bed (e.g., Walder and Fowler, 1994; Collins, 1996; Willis et al., 1996; Swift et al., 2005; Riihmiki et al., 2005; Delaney et al., 2018, 2019; Delaney and Adhikari, 2020), and this applies for both suspended and bed loads (Walder and Fowler, 1994). The residence time of sediments through such processes has yet to be determined, but one may expect it to be substantially shorter than the characteristic time estimated here, as sediment exhaustion is typically observed at a seasonal timescale. Instead, the characteristic time that is derived here corresponds to the glacier characteristic time, as the authors simply track sediments within the glacier. Equations have been developed to estimate such a characteristic time (e.g., Johannesson et al., 1989;Oerlemans, 2001, 2008, 2012; Roe & O'Neal, 2009; Roe et al., 2017; Herman et al., 2018), and all reach the conclusions that the glacier response time is proportional to its length divided by a characteristic velocity. If one takes a length of 15 km and a characteristic velocity of about 10 m/yr for the Tiedemann glacier – here the models show maximum velocities about 75 m/y – one obtains a characteristic time of about 1500 years. Therefore, the results obtained here do not tell us much about the time required to reach an equilibrium for detrital age

distribution, but instead about the glacier dynamics. This is an important shortcoming of the current paper, but I think it could be addressed in a revised version. One possible option is to simply remove the discussions on time scale and solely focus on the shape of the SPDF. A second, more demanding option would be to include a subglacial hydrology model that accounts for sediment transport (e.g., Collins, 1996; Creyts, et al., 2013; Beaud et al., 2018; Delaney et al., 2018).

Second, I have difficulties understanding why the authors have specifically chosen not to apply the model to the existing data (Ehlers et al., 2015; Enkelmann and Ehlers, 2015), which they keep referring to. I think that the approach developed here would have great value and potential to improve our understanding of glacial erosion processes and/or better estimate the contributions from glacial and hillslope erosion, so why not try to fit the model to actual observations, especially when detrital data collected in moraines and glacial outwash are very similar? Our current knowledge about glacial erosion processes, or an erosion rule, is limited to a relationship between sliding velocity and erosion that has limited amount of observational support (Humphrey and Raymond, 1994; Herman et al., 2015; Koppes et al., 2015; Cook et al., 2020), so why not do it here?

Finally, it is surprising that the codes are not made publicly available. ESurf is an open-review and open-access journal. The data policy of the journal states: "In addition, data sets, model code, video supplements, video abstracts, International Geo Sample Numbers, and other digital assets should be linked to the article through DOIs in the assets tab." I could not find any link or doi for the codes. The codes that are used here have been developed for more than 10 years, and are still not available in the public domain.

Specific comments: - Abstract: how does detrital thermochronology enables to avoid biases better than other methods? - 1/22: SPDF should be spelled out the first time it is used. - 2/3: the authors should be more precise on the order of timescales. (see main comments about existing work on characteristic timescales.) - 3/5-17: Detrital studies

can only done appropriately, in my opinion, if the source area is properly described. The fertility or age distribution within the catchment must be characterized as much as possible. It is clearly not the panacea as the problem remains ill-posed, but at the very least the authors could make some references about the importance of having a good knowledge of the source area. - 4/9-10: The code should be made publicly available. See main comments. - 6/11: "We follow MacGregor et al. (2009).." and everybody else (e.g., Braun et al., 1999; MacGregor et al., 2000; Tomkin and Braun, 2002, etc.) - 6/19: While there is some observations that support the link between sliding and erosion (e.g., Humphrey and Raymond, 1994; Herman et al., 2015; Koppes et al., 2015; Cook et al., 2020), there is no available data specifically for the chosen erosion rule, beyond the models of Uglevig et al. (2018). It would be good at least if the authors could acknowledge some of the observational basis for utilizing of this rule, or the relationship between sliding and erosion. - 6/22: I do not think subglacial fluvial transport should be ignored. See main comment. - 7/22: The flux and erosion rate (i.e., velocity) are the same equations (Eqns. 7 and 8) both scaled with constant that have the same unit. That cannot be. - 7/10: The authors assume that all the transport happens within the ice. See main comment on this assumption. - 8: Is there any information on the actual velocity of the glacier? The authors have chosen a relatively slow glacier, although the glacier is comparable to many alpine glaciers and I appreciate that the authors needed a site where some thermochronological data were available. This has some influence on the final result of the characteristic timescale, as it scales as the ratio between the glacier length and velocity. For example, Cook et al. (2020) showed velocities ranging from a few meters per year to several kilometers, implying that the equilibrium timescale estimated here is only applicable to the Tiedemann glacier. - 10/11: It would be useful to have more information about the geology. Ehlers et al. (2015) refer to Rusmore and Woodsworth, but the geological map is very large. Is the geology under the glacier truly uniform? I could not find this information. - 12/4: "limiting the ability" I do not understand why. Intuitively, more variations, such as kink in an age-elevation profile, should provide more information. - 12/14-16: The time to travel through the glacier is

entirely dependent on the ice flow model, and it is likely it would be significantly faster if the subglacial hydrology would be included. - 19/4.1: See main comment about sediment transport. - 19/25: There are numerous papers on the glacier response time that could cited. See main comment.

---

## Editor Comment (EC1) · Jean Braun (Editor) · 20 Apr 2020

The two reviewers have provided extensive reviews of the manuscript and both agree that it contains interesting material that should be published in ESurf. They both note, however, that the method used requires to be better explained. One important point raised by the anonymous reviewer concerns the efficiency of transport by subglacial water flow and how it might affect the authors' conclusion regarding the time it takes for detrital distributions to reach "steady-state". There is a need too, as suggested by T Ehlers, to show how sensitive the results presented here are to the rather arbitrarily

chosen set of model parameters. I also agree with the anonymous reviewer that the manuscript would greatly improve its potential impact if it made a better use of existing data (form the study area) to show its robustness. I strongly encourage the authors to provide us with a response to the two reviewers' comments, based on which I would like to recommend that they prepare a revised version.

---

## Author Comment (AC1) · 3 Jun 2020

**Interactive comment* on "Modelling the effects of ice transport and sediment sources on the form of detrital thermochronological age probability distributions from glacial settings" by Maxime Bernard et al.**

**Response to the reviewer.**

**We first wish to particularly thanks the Pr. Ehlers for his review of the paper. His comments are very relevant and have been very appreciated by the authors.**

**We took into consideration all the comments and we present our reply point by point.**

First, we wish to point out that we changed the title of the manuscript to be more informative about the approach of our study.

GENERAL COMMENTS:

G1. *The general validity (beyond the simulations presented) of the authors results and interpretations is difficult to assess. If I understand the text and Table 1 correctly, the authors present a set of simulations with only one set of parameters. These results are nicely presented and described, but the primary interpretations of the paper (e.g. 1,500 yr equilibrium time for observed age-distributions; sediment trapping in tributary glaciers, hill slope vs. Glacial contributions to observed age distributions) are based on this single (?) set of chosen parameters. The results would be generalisable and more broadly applicable if a small set of additional simulations with a sensitivity analysis were included. For example, picking the least well constrained climate, ice, and hill slope parameters and picking reasonable values above and below what is shown in Table 1 should be considered to evaluate how robust the interpretations are in the text. I suggest these additional figures be shown in the supplementary material and then referred to in the main text, or highlighted in a new discussion section with a new figure that compare results.*

In the initial version of the study we presented the models with only one set of parameters for the simulated glacier dynamics. We share the concern of the Pr. Ehlers, so we performed a set of additional simulations with a set of parameter values that lead to a glacier size ~1000 m longer and shorter than the modelled glacier presented in the main study. We chose this range of glacier size difference because greater discrepancies would lead to very different glaciers dynamics and glacier thickness compared to the ITMIX experiments results, which is our calibration for the Tiedemann glacier. The additional models are now presented in a dedicated section in the supplementary materials, where we compare the results with the reference model (i.e. presented in the main text). Overall, the resulting detrital age distributions and equilibrium time for the frontal moraine of these additional models do not significantly vary from the reference model.

G2. *The results of Enkelmann and Ehlers (2015, Chem Geo) compared AFT ages across the ice cored moraine, outwash, and older moraines in front of the modern glacial terminus. The ice cored moraine and outwash samples were statistical identical (Fig. 6E), meaning that multiple ice cored samples when combined produce the same grain-age distribution as outwash. "slight differences" were observed between the individual ice cored moraine material in this study (Fig. 4 of Enkelmann and Ehlers). While the authors of this study (Bernard et al) explicitly say they are not trying to match observations in their study, in the discussion section they end making statements that do compare / evaluate the observations. There are suggested revisions related to this:*

   *G2-a. In the start of the paper when describing the previous observational studies in the area (and the bedrock data you use), please add text that makes it clearer how these data were collected and which data sets are or are not relevant to how the model is setup and why. For example, the Enkelmann and Ehlers 2015 data are well suited for comparison to the model results (they come from ice cored moraine material at the end of the glacier). In contrast - the Ehlers et al. 2015 data are from*

*glacial outwash and are not suitable for comparison to how the model is setup. The modelling approach does not track particles through the subglacial hydrologic system. No comparisons to this later study should be made in this manuscript.*
*G2-b. The concluding discussion section (4.4) and Fig. 10 make model predictions that are more or less comparable to the observations of Enkelmann and Ehlers, 2015. Although some qualitative comparisons are made in the text, the manuscript would be much stronger if the data from Enkelmann and Ehlers, 2015 were also plotted in Fig. 10c and similarities / discrepancies are discussed.*

Initially, we did not aim to compare our results explicitly to the observations given that our synthetic steady Tiedemann glacier dynamics does not reflect the real Tiedemann glacier which is currently retreating. However, the comments of the two reviewers convinced us to consider such a comparison. As mentioned by the Pr. Ehlers, as our sediment particle sampling is focused on the frontal moraine, we only compare our results with the detrital age distributions from Enkelmann and Ehlers (2015) in which the samples come from the ice-cored terminal moraine. Such a comparison, with statistical tests, is presented in the revised version of the manuscript.
At the start of the paper we clarify the provenance of the samples from Enkelmann and Ehlers (2015) and Ehlers et al. (2015). We have also specified that we produced our bedrock ages according to the age-elevation from Enkelmann and Ehlers (2015) for the AFT data and from Ehlers et al. (2015) for the AHe data. In the revised version of the manuscript, the comparison of our detrital age distributions with the AFT data of the ice-cored terminal moraine samples from Enkelmann and Ehlers (2015) is made in section 4.4 (i.e. Implications for detrital sampling strategies) with the figure and the results from statistical tests presented in the Appendices. In that section we specified the provenance of the samples from Enkelmann and Ehlers (2015) (i.e. ice-cored terminal moraine).

*G3. The model setup and description needs to be clearer. It is not clear how erosion is done for hill slopes and how ages from hill slopes are mixed with the glacial sourced ages. It would greatly improve a readers understanding of this study if the coupling (and particle tracking) between hill slopes and glaciers was better explained, and also the sensitivity of their results (e.g. Fig. 7) to some of these parameters (e.g. see also comment G1 above). I also don't understand exactly how the 'uniform' erosion model was calculated with the ice model (see detailed comment below). The methods sections needs to explain the model setup for all this better. Section 2.1 (end) explains a non-linear diffusion model is used for hill slopes, but it's not described in enough detail how the sediment transport is done in this part of the landscape. The paper later on nicely explores how hill slope vs. Glacial sediment sources impact detrital cooling ages. Thus, some additional text in the methods section would make it much easier to understand the model results. As it's written now, I could not reproduce your results if I wanted to.*

We considered the comment of the Pr. Ehlers and have made the model setup and description clearer having entirely rewritten section 3. In the case of uniform erosion, we specified the production of particles in each cell of the model for both hillslope and glacial sources. In the case of non-uniform erosion (section 3.5), the erosion is determined by the erosion laws presented in section 2.1. The thermochronological age of a particle is based on the source location elevation of that particle and the appropriate age-elevation profile. Thus, each particle carries an age which is not modified during the particle transport. The mixing of particles (i.e. ages) is the result of the transport pattern. However, during the sampling process we can choose to identify and sample particles according to their source location (hillslopes or glaciers). The sensitivity of parameters for figure 7 are now presented in the supplementary materials.
A particle is formed once the erosion products in a cell reach a thickness threshold ($Hs = 0.01$m). Then, each particle is transported away according to the transport laws for hillslopes (Eq. 8) and for ice (Eq. 9). This is now explained in the start of section 2.2.

*G4. Comparison of distributions should include statistical tests. Section 4.4 (see also detailed comment 26 below) presents an analysis of how representative point samples across a moraine compare to the mean of all samples and the bedrock distribution. This section is very useful for understanding how and where people could sample. However, the analysis does not statistically*

*evaluate if the different synthetically sampled distributions are the same or not. Visual / qualitative comparisons of distributions is dangerous, and I suspect (based on experience) that several of the distributions show in Fig. 10c will be statistically identical. If this section remains in the paper (which I hope it does), it's important the authors conduct a simple KS or Kuiper test to see if in fact the different distributions show in figure 10c are different or not. This comment could also potentially impact the conclusions presented in section 4.5. The text in section 4.4 and 4.5 is good, but simply needs support from a more quantitative comparison. Finally, as mentioned above for comment G2, this model result should be compared to observations that were collected from roughly this same area (Enkelmann and Ehlers, 2015).*

In the initial version of the manuscript, our comparative analysis was primarily qualitative as our goal is to describe the form of the detrital age distributions and link it to processes (i.e. sediment transport and erosion). In section 4.4, we compare the detrital age distributions over 4 regions within the frontal moraine to the catchment bedrock age distribution, and discussed the discrepancies that occur. The validity of statistical tests is not a given (e.g. Vermeesch, 2018), partly our concern regarding statistical tests. However, in the revised version of the manuscript we use two statistical tests (Kolmogorov-Smirnov and Kuiper tests) to compare our modelled detrital age distributions with the catchment bedrock age distribution. Given, this, we also highlight some contradictions between the inferences made from the two statistical tests.

*G5. The text is in general well written and clear. However, there are many small wording issues / grammatical problems throughout the text (e.g. missing articles, subject/ verb agreement) that are understandably hard to catch for a non-native speaker. I recommend a native speaker/co-author give the text another thorough read to correct these.*

We thank the Pr. Ehlers to have pointed out such wording and grammatical issues. We brought a particular attention to these issues in the revised version of the manuscript.

**SPECIFIC COMMENTS:**

0. *For clarity - all axis labels with 'Age' on them should say what age (e.g. "AFT Age") you are plotting since you work with two different systems in this study.*

Done.

1. *Abstract should make it clear if the langragian particle tracking is only for ice flow, or subglacial water.*

We now state in the new version of the manuscript (page 1 – line 15): "Sediments are tracked as Lagrangian particles formed by bedrock erosion, where their transport is restricted to ice or hillslope processes until they are deposited".

2. *Page 2 paragraph starting at line 17: Also relevant to the content of this paragraph, and the study area investigated is the study by Yanites and Ehlers 2016 that documents how glacial sliding relates to bedrock thermochronometer ages (in a neighbouring valley in the Coast Mountains). Yanites, B. J. and Ehlers, T. A.: Intermittent glacial sliding velocities explain variations in long-timescale denudation, Earth and Planetary Science Letters, 450, 52–61, doi:10.1016/j.epsl.2016.06.022, 2016. Also - this manuscript is highly relevant to the following previous work and the authors should consider citing it in the introduction or discussion. Herman,F.,et al., 2018. The response time of glacial erosion. JGR - Earth Surface.* [https://doi.org/10.1002/2017JF004586](https://doi.org/10.1002/2017JF004586)

We thank the Pr. Ehlers for suggesting the studies relevant for our manuscript. We have added a small paragraph that mention the results of the study of Yanites and Ehlers 2016 (Page 2 -lines 20-24).

We also now mention the study of Herman et al. (2018) when we discuss the kinematics of our sediment transport model (Section 4.2) and the characteristic timescale for the glacier dynamics (Page 20 – lines 28-29).

3. *Page (pg) 3, line (ln) 12 - It might be worth clarifying here for readers that the Enkelmann and Ehlers 2015 studied sampled ICE across the ablation zone, and the Ehlers et al. 2015 study sampled glacial OUTWASH. This would help readers understand why you can not directly compare model predictions to one of these sets of observations..*

We have made this clear by modifying this sentence. (Page 3 – lines 16-19).

4. *Pg5, 6 model description. Hydrology effects on sliding are described here, but please also add a sentence or two that says if this is also included in the particle tracking for making SPDFs of cooling ages. Maybe you address this later.*

We have added the sentence "We stress again that we neglect the transport of particles by meltwater and focus only on the ice and hillslope processes." (Page 7 – lines 22-23).

5. *Pg5, 6 - It is also important that this section says how you have calibrated the model for the subsurface hydrology. This aspect of the model is likely very important for how the data are interpreted, so some text on this aspect would be useful.*

We have added the sentence "We calibrated our hydrological model by a trial-error process by varying $k_0$ (Eq.2) to lead to a reasonable value of basal ice sliding velocities (Table S1)." We have also added a short description in the supplementary material associated to Table S1 to show the parameters used for our calibration approach for the Tiedemann glacier. (Page 6 – lines 7-8, and Supplementary Table S1).

6. *Pg.6 - Your approach assumes all sediment comes from quarrying. I'm more or less ok with this (note fine sediment fractions were present in what we sampled for this glacier). However, please explain here if you account for the comminution (breaking down) of plucked material. Why could this be important? If 20x20cm rock is plucked in the upper reaches of the catchment it will break down during transport and provide fine grain material that was sampled. If a 20x20 cm rock is plucked from 100 m from the sample point - it wouldn't show up in sample. The material sampled for the Tiedemann glacier ice cored moraine and outwash was a 'bulk' sediment sample, but with nothing greater than _2x2cm size in it. I would be great if your modelling approach accounted for comminution, but I'm guessing this is not the case. So this effect needs to be acknowledged and the potential implications of it discussed in a model caveats section.*

We do not account for the comminution in our transport model as this would lead to computer-memory issues due to the too large number of particles tracked by the model. We have added a short paragraph to discuss the issue of such process in the sampling approach, in section 4.1 which deals with the limitations of our modelling approach (Page 20 – lines 12-15).

7. *Pg 6/7 - section 2.3. As indicated above, please explicitly state that water transport of detritus is not accounted for. Fluvial systems mix sediment very efficiently and the flow rates on these outlet rivers are high (rounded cobbles were in the river bed and appeared transported by it).*

This comment is similar to the specific comment 4 that we have already answered to (Page 7 – lines 22-23). We also remind this issue page 20 – line 16.

8. *Pg. 8 section 2.3. Please add some text saying how the glacial mass balance (and climate inputs) are calculated and refer to your table.*

We have added a Table (S1) in the supplementary materials and we explain how different parameters values have been chosen to calibrate our model. Furthermore, a short text was also added to describe the computation of the glacier mass balance (Supplementary material – Table S1).

9. *Fig. 4c, d - I suggest labelling the x-axis with the age plotted (e.g. "AHe Age" for c). Also - for the caption, mention what uncertainty you used for making the PDFs since this influences the smoothness of the curves relative to panel B. Caption should also explicitly say the data come from glacial outwash, not moraine.*

We have changed the figure label accordingly. However, the data used for the building of bedrock age distributions are the in-situ central ages of the AHe and AFT systems, through the age-elevation profiles, from Ehlers et al. (2015) and Enkelmann and Ehlers (2015) respectively, and thus are not from the glacial outwash sample presented in Ehlers et al. (2015). We describe these bedrock SPDF Page 12 – section 3.1.

10. *Pg12 ln7. Please say under what conditions the equilibrium state was calculated, and how closely it matches the present-day thickness and length of the glacier.*

We explain our calibration approach for the synthetic Tiedemann glacier Page 8 – lines 15-18. We also refer the readers to Figure S3, which shows the comparison between the iSOSIA model and the results of the ITMIX experiments (Farinotti et al., 2016) that predicts the ice thickness of many glacier around the world, as the Tiedemann glacier.

11. *Fig. 5. Please provide a more descriptive caption of what model this is at the start of the caption. Also - what are you actually doing with the 'hillslope' vs. 'glacial' parts of the catchment (Fig. 5a). It's not clear from the text (or caption) if you are also feeding hill slope material into the SPDFs calculated.*

Changes in the caption of Fig. 5 have been made to make it clearer (Figure 5 and Page 12 – line 26). We also added a sentence in the main text: "Particles are sampled independently of their source origin (hillslope vs glacial, Fig. 5a)."

12. *Pg13, ln1-2. Reword sentence please.*

Done. (Page 15 – lines 6-7).

13. *Fig. 6 - caption needs a starting sentence saying in general what is plotted and what model it comes from. Also- again for panel c, d - indicate the age type ("AHe Age") on the x-axis. Please do this for all other figures if this is also the case. It makes it much easier to read the figures quickly.*

Additional sentence and axes labels have been modified according to the comment for all figures.

14. *Pg.14 ln1. Please explain better what you mean by "model with uniform erosion". I'm confused because I don't understand how you made the ice model have uniform erosion - and where the uniform erosion was applied (e.g. Hillslopes and glacial areas?). Fig. 7e kinda gets at this, but the text should explain it better. Thanks.*

We forced uniform erosion by setting the production of particles in each model cell with a constant erosion rate. The erosion is applied in the entire model. The different colours in Figure 7a (i.e. red and blue) identify the sediment sources (hillslopes and glaciers). According to this comment and the following one, section 3.3. was confusing. We have therefore entirely rewritten it to make our results clearer. The model with uniform erosion is now explained more specifically at page 15 – lines16-17.

15. *Pg. 16 top. After reading this page (related to previous comment) I'm still confused and some clarification is needed on this paragraph and what was actually calculated. The start of the*

*paragraph needs to explain better what the objective of this comparison is (hillslope vs. glaciers) . Also ln3-4 are confusing because I thought this section only about uniform erosion, but this sentence says you are comparing a detrital SPDF to the uniform erosion model. Are you talking about the OBSERVED detrital SPDF? Perhaps make it clear throughout the text by always using 'observed' vs. 'modeled' detrital SPDFs*

As mentioned in a previous comment we have rewritten Section 3.3. We compared the modelled detrital SPDF with the modelled catchment bedrock SPDF resulting from the uniform erosion model. We differentiate between the detrital SPDF and the catchment bedrock SPDF clearer.

16. *Pg16, ln10+. This paragraph is also unclear. After reading all of section 3.3 - I'm confused as a reader. Please rewrite this section to make it clearer of a) what is the logic behind the experiment / comparison conducted, b) what is the first and second order main trends in the results and what data / model results (bedrock vs. Detrital you're looking at, and c) summary sentence(s) with the key observation to take away.*

We took into consideration this comment and have rewritten section 3.3. The rationale behind the model of uniform erosion was to characterise the form of the detrital SPDF at the glacier front resulting from a continuous spatially uniform production of particles (which differs from the model presented in Figure 5 that consider only a pulse of particle production). In this case, we should expect some effect due to the transfer time of sediment particles as we can see with the shifting of the mean detrital SPDF toward younger ages (Fig. 7). We discussed this effect in section 4.2.

17. *Maybe I missed it earlier, but after reading the results section - I think the methods section needs to be expanded some to explain how you look at (or calculate) hillslope vs. Glacial contributions to the detrital cooing ages.*

The particles can be sampled according to their source origin (each particle as a tag according to the source when forming that can be retrieved during the sampling process). We now specify this at the end of section 2.4 (Page12 – line 4).

18. *Pg16 ln 29. I don't understand how "we computed a new: : :". How did you compute this? Assuming uniform erosion? Using a diffusion based hill slope transport law? Please elaborate.*

We calculated a new bedrock SPDF using the age-elevation relationship convoluted to the hypsometric distribution of the hillslope sources, as we assume uniform erosion. We now refer to this bedrock SPDF as the "hillslope bedrock SPDF" in the main text. (Page 16 – lines 10-11)

19. *Pg.17 ln5-10. This result is entirely dependent on the assume hill slope transport law used, and diffusivity,: : :.right? Perhaps mention this, and also make it clearer (per my previous comments) how the curve in Fig. 7 is calculated.*

Figure 7 displays results obtained with a uniform erosion model. There is therefore no dependency on the erosion law used for the hillslopes. Concerning a potential sensitivity to a transport law, because we consider steady-state SPDFs, our results are assumed independent of the transport processes or rate. We make this clearer by performing sensitivity tests by varying the diffusivity value used in this study ($K_h = 5$ m² yr$^{-1}$). The results of these tests are presented in the supplementary materials. The mean detrital SPDFs (in the frontal moraine) resulting from particles originating from hillslopes are computed following the method presented in section 2.4. The hillslope bedrock SPDF is computed using the age-elevation relationship and the hypsometric distribution of the hillslope sources.

20. *Pg. 18 ln1. Several times in the paper you refer to or tune the model to an erosion rate 1 mm/yr. Why did you use this value? This should be explained in the methods section.*

We have used a value of 1 mm yr$^{-1}$ as it is in the range of natural values and it allows a continuous production of particles while maintaining a reasonable simulation time. We have added these sentences at the top of section 3.3 (Page 15 – lines17-18).

*21. Pg.18 ln1-4. I don't understand this sentence and where these numbers you cite are coming from. For example, where is 31 mm/yr coming from? Where are the uncertainties in the numbers later in the sentence coming from?*

We meant that despite the mean erosion rate is 1 mm yr$^{-1}$, local deviations to this average value occur due to the heterogenous pattern of erosion. The value of 31 mm yr$^{-1}$ is the maximum local erosion rate shown in Fig. 9 (in the revised version of the manuscript). Next, we calculated the standard deviation for each sediment sources (hillslope vs glacial) to the average erosion rate of 1 mm yr$^{-1}$. We clarify this in the revised version of the manuscript. (Page 18 – lines 17-19).

*22. Pg. 18. Ln6. Please rewrite this sentence. I don't understand it.*

The sentence has been rephrased and split into 3 sentences (Page 18 – lines 19-21)

*23. Pg. 19 ln1. "traducing"? Not sure what you're trying to say.*

This sentence has been changed to clarify this.

*23. Pg 19 ln20. Please expand this thought (particle tracking is in ice, not sub-glacial drainage). I would describe in general (qualitative) terms how the results could differ if subglacial outwash was sampled (e.g. how Ehlers et al. 2015 sampled).*

We have done so, see page 20 -lines 16-21.

*24. Pg. 19. Ln 24-25 "spatial erosion pattern can be biased on the detrital SPDF (e.g. Ehlers et al. 2015)". Please remove the "e.g. Ehlers et al. 2015". You may be correct that there is a bias, but you haven't shown this because the Ehlers et al. samples were from OUTWASH, not from ice. Also - your timescale arguments for bias later in the paragraph would likely be severely decreased if outwash is sampled because transit times of water to the outlet are significantly faster than for ice. So, please either show that Ehlers et al. 2015 have a bias in their outwash sample interpretation, or remove the reference to this paper to be fair. Final note concerning the general conclusions you are trying to make here about timescales - while your description is accurate for the simulations you present, it's hard to know if this is really a general result with any sensitivity tests to your model parameters presented in the study. Please consider adding sensitivity tests in the supplemental material and referring to them in the text. Table 1 - please be more specific about what you mean by "variable" in some rows. I think what you mean is that these variables are 'internally calculated'*

The reference has been removed. The sensitivity tests have been added in supplementary materials, section 2. They show that the time to reach equilibrium, neglecting fluvial sediment transport, is of the same order than presented in the text ($10^3$ years), when varying the model parameters. Obviously, considering fluvial transport would decrease drastically the response time of sediment transport. Yet, because we are only considering the detrital signature of moraines, which are glacial sediment deposits and not fluvial ones, we do not expect a significant impact of considering fluvial transport on our results. We have revised the scope of our conclusion to account for this important point (Page 20 – lines 28-32 and Page 21 lines 1-11). Moreover, Table 1 has been modified according to the comment: "Variable" is replaced by "Model outcome".

*25. Pg. 21 ln29-30. This isn't really a surprise is it that moraines will more closely represent glacial erosion than hill slopes? Also - doesn't this statement and paragraph sort of contradict the previous section where you say there would be a bias towards low elevations in SPDFs? Maybe*

*I'm missing something here - but perhaps more text (in the previous section about biases) relating to what section 4.3 is saying would help clarify things more.*

In the case of debris-cover glaciers where the major sediment sources are hillslopes, thus moraines likely represent hillslopes erosion. We want to point out here that because glacial sediment particles are likely to be produced close to the ice streamline showing high velocities, the transfer time of such particles will be small compared to supraglacial sediment particle. Thus, in the case where the amount of erosion is the same for supraglacial and subglacial sources, we expect that the glacial sources to be over-represented, and thus the detrital SPDF to be bias toward younger ages if they are located beneath the ice (Page 23 – lines 13-16).

*26. Pg.22 section 4.4. This is nice that you done this analysis. However, to do this type of comparison you need to test if the distributions show in Fig. 10c are in fact statistically different. To do this, you need to apply a KS (or Kuiper) test of the distributions.*
*Please include this type of analysis to see if these distributions really are in fact different at the 95% confidence level. This comment also potentially impacts the interpretations presented in section 4.5.*

We already reply in the general comment G4.

*27. Pg. 24 ln 1-2. This conclusion is correct for the simulations presented, but how variable is this result if there are variations in some of the model parameters. A sensitivity test of model inputs would help readers see if this result is general, or specific to your model simulations.*

We have made such tests and discussed the results in section 2 of the supplementary materials. The results do not significantly differ from those presented in the main text.

*28. Pg. 24 ln22-23 "However, we also emphasis ⠒ ⠒ ⠒" This statement should be removed or significantly expanded & justified. How many grains should be sampled is a significant topic on it's own, and this study doesn't address this. The typical "100" grains minimum that people site from Vermeesch is not actually correct for these types of detrital samples (we have a paper in preparation that goes through the statistics of number of detrital AHe samples needed for different catchment sizes and it's complicated).*

We removed this sentence from the conclusion.

*29. Supplementary materials. There is useful material presented in the supplement, but the figure captions are a too sparse for some figures. Please expand the figure captions some to be more descriptive of what is shown. For example, what model simulation is shown in the figure, and perhaps also what section of the text the figure is relevant to.*

The captions of the figures have been rewritten, expanded and clarified for better description of the figures.

**Reference**

Vermeesch, P.: Dissimilarity measures in detrital geochronology. *Earth-Science Reviews*, *178*, 310-321, 2018.

**Code availability**

The version of iSOSIA (iSOSIA_3.4.7b) used for this study, as well as the external routine to compute the detrital age distributions, are now publicly available at: https://github.com/davidlundbek/iSOSIA.

---

## Author Comment (AC2) · 3 Jun 2020

**Interactive comment* on "Modelling the effects of ice transport and sediment sources on the form of detrital thermochronological age probability distributions from glacial settings" by Maxime Bernard et al.**

**Response to the reviewer.**

**We thank the anonymous reviewer for his review of the manuscript. We here response to the main concern of the anonymous reviewer and then we provide a point by point response to his comments.**

We first point out that we changed the title of the manuscript to be more informative about the approach of our study.

The first concern is about the limited sediment transport processes considered in our study. We considered the transport of sediments on hillslopes and by ice but do not incorporate transport by meltwater. The anonymous reviewer pointed out that the majority of sediments is transported out of the glacier through the subglacial hydrology system. This may have important implications concerning the sediment transfer time and thus on the equilibrium time of the frontal moraine reported in our study (about 1500 years). We understand and share the concern of the anonymous reviewer about the role of subglacial hydrological systems in reducing the transfer time of sediments. We have revised the scope of our conclusions to discuss this important point. However, as we only consider the detrital signature of moraines, which are glacial sediment deposits and not fluvial ones, we do not expect a significant impact of fluvial transport on our results. We also discuss this point further in the revised version of the manuscript in section 4.2.

Our time for equilibrium of the frontal moraine (i.e. 1500 years) is similar to the characteristic time estimated by some authors the anonymous reviewer has cited. However, as mentioned earlier, we expect that the frontal moraine mainly reflects the glacier dynamics as the sediments that participate to build such glacial features mainly come from the ice (e.g. Winkler and Matthews, 2010; Bowman et al., 2018; Ewertowski and Tomczyk, 2020). Moreover, our discussion includes a spatial distribution on sediment transfer times which is not captured by the equation estimating the characteristic time. For these reasons we limited our discussion to sediment transfer times but revised the scope of our conclusion.

Initially, we did not aim to compare our results explicitly to the observations given that our synthetic steady Tiedemann glacier dynamics does not reflect the real Tiedemann glacier which is currently retreating. However, the comments of the two reviewers convinced us to consider such comparison. We now compare our modelled detrital AFT distributions to the detrital AFT distributions from Enkelmann and Ehlers (2015) which are coming from ice-cored terminal moraine (see Section 4.4).

We are aware that the erosion rule of Ugelvig et al. (2016) is mainly based on a mechanical model for bedrock fractures (Iverson, 2012), which has not been validated by a comparison with field data. On the other hand, empirical models that consider power laws between erosion rates and sliding velocity obtained by fitting natural data have little physical support and may lack some important processes. For instance, it is well documented that effective pressure plays a role in the quarrying process (e.g. Cohen et al., 2006). The equation of Ugelvig et al. (2016) is an attempt to incorporate this effect on a large-scale model. As discussed in Ugelvig et al. (2016), the strong dependency of erosion to the effective pressure (the power of 3) may be exaggerated. However, the results presented in Fig. 9, do not contradict the observations. If we consider the mean catchment erosion rate (1 mm yr$^{-1}$) and the maximum sliding speed of the main glacier ($\sim$60 m yr$^{-1}$, Fig. 3), we are in the range of values

presented in Cook et al. (2020). Overall, we chose this model because of its mechanistic basis, despite its lack of a validation by natural data.

We share the concern of the anonymous reviewer about the availability of the source code. Therefore, we now make the code for iSOSIA and the external routine for constructing age distributions publicly available. (https://github.com/davidlundbek/iSOSIA).

**Specific comments:**

1. *Abstract: How does detrital thermochronology enables to avoid biases better than other methods?*

We did not make a comparison to other methods, but meant that thermochronological analysis on sediments (detrital thermochronology) potentially provides information from beneath the glacier, contrary to in-situ thermochronology for which sampling beneath the ice is generally not possible. (Page 1 – line 10).

2. *Page 1-line 22: SPDF should be spelled out the first time it is used.*

Done.

3. *Page 2 – line 3: The authors should be more precise on the order of timescales.*

This point has been removed from the abstract.

4. *Page3 – lines 5-17: Detrital studies can only be done appropriately, in my opinion, if the source area is properly described. The fertility or age distribution within the catchment must be characterized as much as possible. It is clearly not the panacea as the problem remains ill-posed, but at the very least the authors could make some references about the importance of having a good knowledge of the source area.*

We share the concern of the anonymous reviewer. Therefore, we have added a small paragraph in the introduction to mention the need of a priori knowledges on the spatial distribution of ages in the catchment and mineral fertility, to interpret the detrital thermochronology data (Page 3 – line 10-11).

5. *Page 4 – lines 9-10: The code should be made publicly available.*

The iSOSIA version to run models presented in this study, and the Matlab code to compute age distributions, is now publicly available.

6. *Page 6 – line 11: "We follow MacGregor et al. (2009).." and everybody else (e.g. Braun et al., 1999; MacGregor et al. 2000; Tomkin and Braun, 2002, etc.)*

We are aware that other authors considered the same assumption about ignoring the abrasion erosion, however to limit the number of references used in this study we now limit our citation to MacGregor et al. (2009).

7. *Page 6 – line 19: While there are some observations that support the link between sliding and erosion (e.g. Humphrey and Raymond, 1994; Herman et al., 2015; Koppes et al., 2015; Cook et al., 2020), there is no available data specifically for the chosen erosion rule, beyond the models of Ugelvig et al. (2018). It would be good at least if the authors could acknowledge some of the observational basis for utilizing of this rule, or the relationship between sliding and erosion.*

We already answered this comment in the responses to the main comments.

8. *Page 6 – line 22: I do not think subglacial fluvial transport should be ignored.*

Indeed, we fully agree that subglacial fluvial transport should be considered in future studies. However, the role of subglacial fluvial transport in the building of frontal moraines may be not predominant. We discuss this point further in the section 4.2 of the new version of this study.

9. *Page 7 – line 22: The flux and erosion rate (i.e., velocity) are the same equations (Eqns. 7 and 8) both scale with constant that have the same unit. That cannot be.*

We thank the reviewer to have pointed out this mistake. This has been corrected in the new version of the manuscript.

10. *Page 7 – line 10: The authors assume that all the transport happens within the ice.*

We have answered this comment in the responses of the general comments.

11. *Page 8: Is there any information on the actual velocity of the glacier? The authors have chosen a relatively slow glacier, although the glacier is comparable to many alpine glaciers and I appreciate that the authors need a site where some thermochronological data were available. This has some influence on the final result of the characteristic timescale, as it scales as the ratio between the glacier length and velocity. For example, Cook et al. (2020) showed velocities ranging from a few meters per year to several kilometres, implying that the equilibrium timescale estimated here is only applicable to the Tiedeman glacier.*

To our knowledge, the only information on the velocity of the glacier comes from the ITMIX experiments (Farinotti et al., 2016, 2019). We share the concern of the reviewer about the characteristic timescale; however, we stress that showing the spatial distribution of sediment transfer times brings additional information relative to a single characteristic timescale, that may hide large variability. We also agree that the variability from glacier to glacier (Cook et al., 2020) may restrict the generality of the conclusion about the equilibrium timescale to the Tiedemann glacier, but we think of this conclusion more as an insight about the equilibrium timescale of the frontal moraine, and it relevance to the interpretation of detrital SPDFs. Future studies should explore this issue.

12. *Page 10 – line 11: It would be useful to have more information about the geology. Ehlers et al. (2015) refer to Rusmore and Woodsworth, but the geological map is very large. Is the geology under the glacier truly uniform? I could not find this information.*

The more recent and precise geological map of the Tiedemann glacier area is from Cui et al. (2017) and is available here. The main lithologies outcropping in the Tiedemann glacier catchments are granodiorite and orthogneiss as mentioned in Ehlers et al. (2015), and suggest low bias in fertility of apatite in the area.

13. *Page 12 - line 4: 'limiting the ability'' I do not understand why. Intuitively, more variations, such as kink in an age-elevation profile should provide more information.*

A kink in an age-elevation profile give more information in terms of exhumation history. However, the ability to track the source of sediments depends on the uniqueness of the age-elevation relationship (i.e. one age = one elevation). For the AFT age-elevation profile, this is not the case as the same age can be interpreted as two very different elevation.

14. *Page 12 - lines 14-16: The time to travel through the glacier is entirely dependent on the ice flow model, and it is likely it would be significantly faster if the subglacial hydrology would be included.*

We agree that transport by meltwaters would decrease the transfer time of sediments to the glacier margin and should be integrated in future studies. However, we postulate that the debris composing frontal moraine are mainly the reflect of the ice dynamics, as their formation is mainly done through dumping of sediment from the ice surface, and bulldozing process (see section 4.2).

*15. Page 19 – line 15: There are numerous papers on the glacier response time that could be cited.*

We included references on the glacier response time in the new version of the manuscript, including Johannesson et al. (1989); Oerlemans (2001) Roe and 0'Neal (2009); Herman et al. (2018).

References:

Benn, D., and Evans, D. J.: Glaciers and glaciation. Routledge, second edition, New York, USA, 2013.

Bowman, D., Eyles, C. H., Narro-Pérez, R., & Vargas, R.: Sedimentology and Structure of the Lake Palcacocha Laterofrontal Moraine Complex in the Cordillera Blanca, Peru. Revista de Glaciares y Ecosistemas de Montaña, (5), 16-16, 2018.

Cook, S. J., Swift, D. A., Kirkbride, M. P., Knight, P. G., and Waller, R. I.: The empirical basis for modelling glacial erosion rates. Nature communications, 11(1), 1-7, 2020.

Cui, Y., Miller, D., Schiarizza, P., and Diakow, L.J.: British Columbia digital geology. British Columbia Ministry of Energy, Mines and Petroleum Resources, British Columbia Geological Survey Open File 2017-8, 9p. Data version 2019-12-19, 2017.

Ewertowski, M. W., and Tomczyk, A. M., Reactivation of temporarily stabilized ice-cored moraines in front of polythermal glaciers: Gravitational mass movements as the most important geomorphological agents for the redistribution of sediments (a case study from Ebbabreen and Ragnarbreen, Svalbard). Geomorphology, 350, 106952, 2020.

Farinotti, D., Brinkerhoff, D., Clarke, G. K., Fürst, J. J., Frey, H., Gantayat, P., ... and Linsbauer, A.: How accurate are estimates of glacier ice thickness? Results from ITMIX, the Ice Thickness Models Intercomparison eXperiment. The Cryosphere Discussions, 2016.

Farinotti, D., Huss, M., Fürst, J. J., Landmann, J., Machguth, H., Maussion, F., and Pandit, A.: A consensus estimates for the ice thickness distribution of all glaciers on Earth. Nature Geoscience, 12(3), 168, 2019.

Herman, F., Braun, J., Deal, E., and Prasicek, G.: The response time of glacial erosion. Journal of Geophysical Research: Earth Surface, 123(4), 801-817, 2018.

Jóhannesson, T., Raymond, C. F., and Waddington, E. D.: A simple method for determining the response time of glaciers. In Glacier fluctuations and climatic change (pp. 343-352). Springer, Dordrecht, 1989.

Oerlemans, J.: Glaciers and climate change. CRC Press, 2001.

Roe, G. H., and O'Neal, M. A.: The response of glaciers to intrinsic climate variability: observations and models of late-Holocene variations in the Pacific Northwest. Journal of Glaciology, *55*(193), 839-854, 2009.

Ugelvig, S. V., Egholm, D. L., and Iverson, N. R.: Glacial landscape evolution by subglacial quarrying: A multiscale computational approach. Journal of Geophysical Research: Earth Surface, 121(11), 2042-2068, 2016.

Winkler, S., and Matthews, J. A.: Observations on terminal moraine-ridge formation during recent advances of southern Norwegian glaciers. Geomorphology, 116(1-2), 87-106, 2010.

---

## Author Response (AR2)

**Final response* on "Modelling the effects of ice transport and sediment sources on the form of detrital thermochronological age probability distributions from glacial settings" by Maxime Bernard et al.**

We thank the two reviewers and the editors for their feedbacks and final reviews. We address here, a response to the general and technical comments of the two reviewers.

Response to first anonymous referee

First, we thank the anonymous reviewer for highlighting the study from Cogez et al. (2018) that we now consider in the final version of the manuscript (Section 4.2), by adding a sentence:" *Terminal moraines of debris-covered glaciers are thus mainly built by a process of dumping debris from the ice surface, and thus reflect glacier dynamics, especially if the glacier remains stable for a long period of time (Sharp, 1984; Lukas, 2005; Hambrey et al., 2008; Benn and Evans, 2013). Moreover, some sedimentological studies on terminal and lateral moraines have shown limited amounts of glaciofluvial-related facies (e.g. Winkler and Matthews, 2010; Bowman et al., 2018; Ewertowski and Tomczyk, 2020), as it is the case for the Tiedemann glacier (Menounos et al. 2013).* **_Moreover, a recent study showed sediment transfer times relevant to building moraines of the order of 100-200 kyr (Cogez et al., 2018)_**. *Therefore, the timescale for building the frontal moraine would strongly depend on (i) the availability of subglacial debris vs supraglacial debris, and (ii) the glacier dynamics*.".

About the minor comments:

1) We do not address the effect of subglacial hydrology in the abstract as we highlight the results from our models that do not include transport of sediments by such a process. However, we now precise in the abstract that we neglect the sediment transport by subglacial hydrology.
2) We thank the anonymous reviewer for having pointed out this mistake and have changed it accordingly.
3) We now specify in the main text the order of magnitude of the "reasonable sliding velocities" which is x10 m yr$^{-1}$, and often observed for alpine glaciers (Benn and Evans, 2013)
4) According to Farinotti et al. (2019), the thickness of the Tiedemann glacier has been estimated through the contribution of three numerical models (i.e. Fig 1, region 02 in Farinotti et al., 2019). These models are from Huss and Farinotti (2012), Frey et al. (2014), and Maussion et al. (2018). The final glacier thickness estimation has been inferred by applying weights to each of these models, 35 %, 29 %, and 35 % respectively (i.e. Table S1, Supplementary information, Farinotti et al., 2019). Each of these model predictions has been evaluated with ice thickness measurements of several glaciers around the world in the ITMIX experiment (Farinotti et al., 2017). Thus, the average deviations between modelled ice thicknesses and observations were respectively 21%, 34 %, and 34%. From these values, we can therefore evaluate the reliability of the ice-thickness estimate for the Tiedemann glacier of about 21*0.35 + 34*0.29 + 34*0.35 = 29 %.
5) We added constant relative noise for AHe ages (i.e. 10 %) and random binomial noise for the AFT ages (i.e. >30 %).

Response to second referee Pr. Ehlers

Pr. Ehlers addressed general comments, and suggested clarifying the bias towards mid-altitude ages mentioned in the study. In our mean detrital SPDFs inferred from the collection of sediment particles across the frontal moraine, we systematically observe an under-representation of mid-ages (e.g. around 6 Ma for the AHe ages). The identification of the upstream sediment sources in the frontal moraine (Figure 8) and the computation of average transfer time of particles from their source to the frontal moraine (Figure 10), both suggest that hillslope sources are more subject to storage leading to large delays of transfer times. This arises because 1) on average the travel distance for the sources are higher than for the glacial sources (see main text, Sect. 4.2), and 2) because some hillslope sources feed tributary glaciers that shows very low sliding velocities (< 1 m yr$^{-1}$).

Whether this behaviour is applicable to all glaciers is a question that our modelling approach results cannot address. Indeed, we only consider a steady-state glacier and neglect subglacial hydrology. Variations of these two factors, can lead to large variation of sediment transfer times.

Cogez et al., (2018) found long transfer times (i.e. of the order of 100-200 kyrs) for sediments currently resting in frontal moraines around Lago Buenos Aires (Argentine), that they link to the intermediate storage in reservoirs (i.e. lakes or subglacial basins). Yet, Guillon et al. (2015) show large differences in source contributions (i.e. periglacial and subglacial) between two subglacial streams of the bossons glacier (Mont Blanc, France). The authors link these differences to the presence of supraglacial channels and sinkholes that helps transfer sediment from the ice surface to the subglacial drainage network.

To clarify, the bias of mid-ages in our modelling results, we added sentences in:

[revised manuscript text omitted]
 1 | $1 / 1.16 \times 10^{-5}$ | $1 / 4.10 \times 10^{-7}$ | $1 / 1.37 \times 10^{-6}$ | $1 / 3.25.10^{-8}$ |
| Region 2 | **0 / 0.057** | **0 / 0.26** | **0 / 0.09** | **1 / 0,01** |
| Region 3 | $1 / 1.63 \times 10^{-9}$ | $1 / 4.80 \times 10^{-8}$ | **1** / $6.42 \times 10^{-22}$ | $1 / 3.26 \times 10^{-24}$ |
| Region 4 | $1 / 4.46 \times 10^{-11}$ | **1** / $4.68 \times 10^{-9}$ | $1 / 5.78 \times 10^{-20}$ | $1 / 3.38 \times 10^{-22}$ |
| Frontal Moraine | **0 / 0.50** | **0 / 0.55** | 0 / 0,87 | 0 / 0,31 |
| S10 | - | - | $1 / 8.83 \times 10^{-16}$ | $1 / 1.30 \times 10^{-17}$ |
| S11 | - | - | $1 / 4.99 \times 10^{-9}$ | $1 / 4.03 \times 10^{-7}$ |

[revised manuscript text omitted]